# The discovery of three-dimensional Van Hove singularity

Wenbin Wu[1,2,3,12], Zeping Shi[1,12], Mykhaylo Ozerov[4], Yuhan Du[1], Yuxiang Wang[5], Xiao-Sheng Ni[6], Xianghao Meng[1], Xiangyu Jiang[1], Guangyi Wang[1], Congming Hao[1], Xinyi Wang[1], Pengcheng Zhang[1], Chunhui Pan[7], Haifeng Pan[1], Zhenrong Sun[1], Run Yang[8], Yang Xu[2], Yusheng Hou[6], Zhongbo Yan[6], Cheng Zhang[5,9], Hai-Zhou Lu[10], Junhao Chu[2,11] & Xiang Yuan[1,2,3] ✉

Arising from the extreme/saddle point in electronic bands, Van Hove singularity (VHS) manifests divergent density of states (DOS) and induces various new states of matter such as unconventional superconductivity. VHS is believed to exist in one and two dimensions, but rarely found in three dimension (3D). Here, we report the discovery of 3D VHS in a topological magnet $EuCd_2As_2$ by magneto-infrared spectroscopy. External magnetic fields effectively control the exchange interaction in $EuCd_2As_2$, and shift 3D Weyl bands continuously, leading to the modification of Fermi velocity and energy dispersion. Above the critical field, the 3D VHS forms and is evidenced by the abrupt emergence of inter-band transitions, which can be quantitatively described by the minimal model of Weyl semimetals. Three additional optical transitions are further predicted theoretically and verified in magneto-near-infrared spectra. Our results pave the way to exploring VHS in 3D systems and uncovering the coordination between electronic correlation and the topological phase.

Van Hove singularity (VHS) originates from the critical point in the electronic band structure with divergence in the density of states (DOS)[1]. The presence of VHS potentially gives rise to exotic phenomena. Unconventional superconductivity[2–5], magnetism[6–8], and charge density wave[9,10] are frequently discovered in systems with VHS near the Fermi level, such as Kagome lattice[10–13] and Moiré superlattice[14–17].

As defined by Léon Van Hove[1], VHS requires being the critical point of the electronic bands along each dimension. For quadratic energy bands in one dimension (1D), band extrema serves as the 1D VHS, which is divergent in DOS following $1/\sqrt{\varepsilon}$ with energy $\varepsilon$ (e.g., Landau band under strong magnetic fields). For two-dimensional (2D) systems or quasi-2D bands in the bulk systems, the DOS of 2D VHS exhibits logarithmic divergence at the saddle points while the band extrema becomes non-divergent. For instance, the 2D VHS is found at the saddle points in the strontium ruthenium oxide[18,19] and layered vanadium antimonides $AV_3Sb_5$ (A = K, Rb, Cs)[20,21]. The 2D VHS in

[1]State Key Laboratory of Precision Spectroscopy, East China Normal University, 200241 Shanghai, China. [2]Key Laboratory of Polar Materials and Devices, Ministry of Education, School of Physics and Electronic Science, East China Normal University, 200241 Shanghai, China. [3]Shanghai Center of Brain-Inspired Intelligent Materials and Devices, East China Normal University, 200241 Shanghai, China. [4]National High Magnetic Field Laboratory, Florida State University, Tallahassee, FL 32310, USA. [5]State Key Laboratory of Surface Physics and Institute for Nanoelectronic Devices and Quantum Computing, Fudan University, 200433 Shanghai, China. [6]Guangdong Provincial Key Laboratory of Magnetoelectric Physics and Devices, School of Physics, Sun Yat-Sen University, 510275 Guangzhou, China. [7]Multifunctional Platform for Innovation Precision Machining Center, East China Normal University, 200241 Shanghai, China. [8]Key Laboratory of Quantum Materials and Devices of Ministry of Education, School of Physics, Southeast University, 211189 Nanjing, China. [9]Zhangjiang Fudan International Innovation Center, Fudan University, 201210 Shanghai, China. [10]Shenzhen Institute for Quantum Science and Engineering and Department of Physics, Southern University of Science and Technology (SUSTech), 518055 Shenzhen, China. [11]Institute of Optoelectronics, Fudan University, 200438 Shanghai, China. [12]These authors contributed equally: Wenbin Wu, Zeping Shi. ✉e-mail: xyuan@lps.ecnu.edu.cn

twisted bilayer van der Waals material is also found to be closely related to the superconducting and Mott-insulating phases[15,22]. However, VHS is generally thought to be absent in 3D cases[1,23]. The classification, notation, and divergence characteristics of band critical points in different dimensions are summarized in Supplementary Section II.

Realization of VHS in more general and applicable 3D systems is desired[24–28] but rarely reported. Here, we propose to realize 3D VHS utilizing the magnetic Weyl semimetal. The proposed physics can be described by the two-band minimal model of Weyl semimetal[29]

$$H_0 = \left(\Delta - m\mathbf{k}^2\right)\sigma_z + v_{xy}\left(k_x\sigma_x + k_y\sigma_y\right), \tag{1}$$

where $\sigma_{x,y,z}$ represent the Pauli matrices; $\Delta, m, v_{xy}$ are material-dependent band parameters with $\hbar = 1$; the 3D vector $\mathbf{k}$ refers to the momentum. As illustrated in Fig. 1, the energy bands are given by $E = \pm\sqrt{\left(\Delta - m\mathbf{k}^2\right)^2 + v_{xy}^2(k_x^2 + k_y^2)}$, which generates a pair of Weyl nodes at the momentum position $(0, 0, \pm k_c)$ with $k_c \equiv \sqrt{\Delta/m}$, if $\Delta \cdot m > 0$. Critical points can be found at zero momentum with the energy of $\pm\Delta$. The presence of Weyl nodes entails the breaking of time-reversal symmetry (corresponding to magnetic Weyl semimetal)[30–38] or inversion symmetry[39–44], which results in the lifted spin degeneracy. By assuming the isotropic in-plane property, $v_{xy}$ and $v_z = 2mk_c = 2\sqrt{\Delta \cdot m}$, respectively, correspond to the in-plane and out-of-plane Fermi velocity of the Weyl nodes (the slope of the Weyl cone near the charge neutral point). If parameter $\Delta$ is raised while other band parameters are fixed, $v_z$ is effectively increased. For small $v_z$, the saddle point at zero momentum is the only critical point (yellow dot) that is non-divergent (manifesting as a finite kink in the calculated DOS spectrum). If increasing $v_z$, the corresponding DOS rises but does not vary qualitatively. After reaching the condition $v_z \geq \sqrt{2}v_{xy}$, a 3D VHS (green dot) appears at the finite in-plane momentum with Mexican-hat in-plane dispersion. Flat dispersion along the tangential direction effectively reduces the dimension of the system. Consequently, this type of VHS manifests as a rare case of logarithmically divergent DOS in 3D, as shown in the calculated DOS spectrum (lower panel of Fig. 1). Meanwhile, the critical point at zero momentum transforms from the saddle point to the band extremum and keeps non-divergent in the

DOS spectrum. The definition of the green dots (3D VHS) in the figures remains the same throughout the article, and so does that of yellow dots (non-divergent critical point).

The tunability of $\Delta$ serves as the prerequisite for realizing the proposed model. It is challenging for the inversion-symmetry-breaking Weyl semimetal since the model parameter is fixed for the given crystal. In comparison, the proposed physics is more feasible for magnetic Weyl semimetals. They are accompanied by the interplay among magnetism, topological phase, and Weyl quasi-particle, which leads to exotic phenomena such as chiral anomaly[45], magneto-optical response[46,47], large anomalous Hall effect[48,49] and Nernst effect[50,51]. In the magnetic Weyl semimetal, the parameter $\Delta$ is sensitive to the external magnetic field, and the exchange interaction between the itinerant electrons and local magnetic moments leads to the shift of the Weyl band by much larger energy compared to the Zeeman energy. This energy shift is also known as "exchange splitting". To account for this effect, we introduce the exchange interaction term[52] $H_{\text{exc}}$ in the Hamiltonian. By applying the external magnetic field $B$ and changing the magnetic structure, exchange splitting can be effectively controlled. Hence, the overall Hamiltonian $H = H_0 + H_{\text{exc}}(B)$ predicts the appearance of the 3D VHS at the critical magnetic field $B_c$ as exhibited in Fig. 1 (more details of the model are given in the "Methods" section). The value of $B_c$ can be estimated by solving $v_z = \sqrt{2}v_{xy}$ and measuring the field-dependent magnetization (refer to Supplementary Section VI). Thus, the discussed exchange interaction offers a promising mechanism to experimentally realize the 3D VHS. The proposed physics might be detected by magneto-infrared spectroscopy, an effective optical technique to probe the evolution of the electronic states under the magnetic field[53–58].

Here we report the evidence of a 3D VHS in the topological magnet EuCd$_2$As$_2$. Owing to the canting of magnetic moments induced by external magnetic fields, the corresponding exchange interaction yields the energy shift of the Weyl bands, as observed in the magneto-infrared spectrum. The abrupt enhancement in the intensity of inter-band transitions is observed at the critical field $B_c \approx 0.6$ T, evidencing the formation of 3D VHS, while the magnetization varies continuously around the critical field. The energy, DOS and even the presence or absence of the VHS are found controllable by the magnetic field. The experimental optical features and 3D VHS agree well with the two-band minimal model of Weyl semimetal. Based on the band parameters

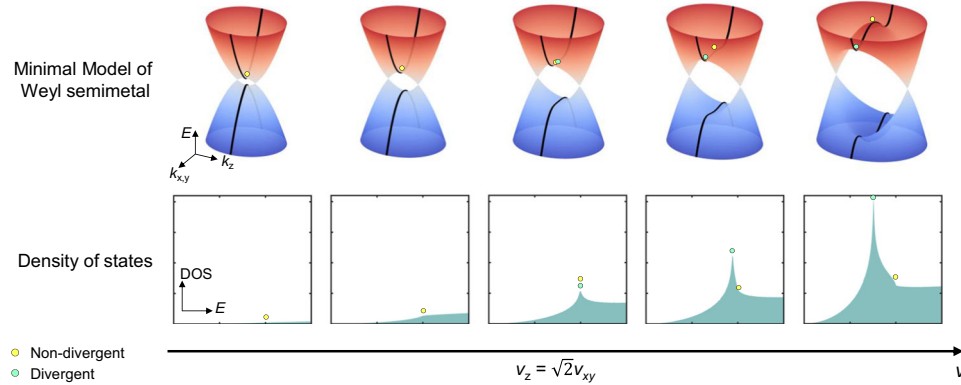

**Fig. 1 | Band evolution and the formation of 3D VHS in magnetic Weyl semimetals.** Considering a two-band minimal model of Weyl semimetals, a tunable band parameter $\Delta$ effectively modifies the out-of-plane Fermi velocity $v_z$. The black curves in the top panel present the calculated energy dispersion along $k_{x,y}$ with $k_z = 0$. For $v_z < \sqrt{2}v_{xy}$, a critical point is located at zero momentum (yellow dot) and shifts with $\Delta$. For $v_z \geq \sqrt{2}v_{xy}$, a VHS appears at the saddle point with finite in-plane momentum (green dot). Such a model can be realized in the 3D magnetic Weyl semimetal. By applying the external magnetic field, one can effectively control the exchange interaction between itinerant Weyl electrons and local magnetic

moments. As $\Delta$ and $v_z$ increasing, the corresponding calculated DOS at the bottom panels exhibits the distinct behavior of these two critical points. The critical point at zero momentum is non-divergent, but that at finite momentum manifests as a rare case of 3D VHS. The energy, momentum, DOS and appearance of the proposed VHS are tunable with external magnetic fields. The controllable formation of VHS at the critical field helps to exclude some possible extrinsic mechanisms in the spectroscopic study. The $E$-axis of the DOS plots is normalized by the energy of the non-divergent critical point for clarification.

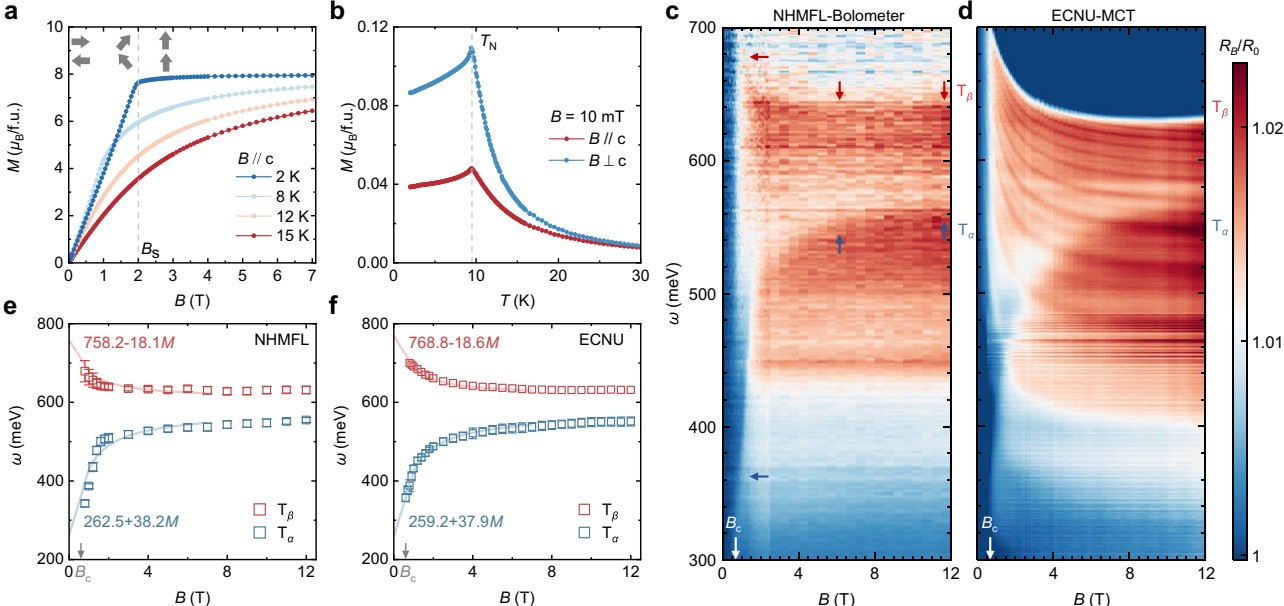

**Fig. 2 | Magnetization measurement and magneto-infrared spectroscopy of EuCd₂As₂.** **a** Out-of-plane magnetic field dependence of magnetization at different temperatures. Denoted by the gray arrows, the magnetic ground state of EuCd₂As₂ is A-type AFM. When an out-of-plane magnetic field is applied, it undergoes spin canting and reaches magnetic saturation around 2 T at 2 K (dashed gray line). **b** Temperature evolution of magnetization indicating an antiferromagnetic transition. **c** and **d** False-color plot of relative reflectivity $R_B/R_0$ of EuCd₂As₂ at 8 K measured in NHMFL and ECNU, respectively. The intensity of optical features increases abruptly at the critical field $B_c \approx 0.6$ T (white arrow) in the spectrum from both setups. The blue and red arrows point out the energy evolution of $T_\alpha$ and $T_\beta$,

respectively. **e** and **f** The comparison between magnetization and the energy of $T_\alpha$, $T_\beta$. The blue and red squares denote the apparent energy of $T_\alpha$ and $T_\beta$. The energy error of $T_\beta$ determined from the ECNU spectra is much smaller than the symbol size. The solid curves represent the linear scaling of magnetization with the scaling parameter labeled near the corresponding curves. It suggests that the continuous shift of the electronic bands originates from magnetization. Meanwhile, the magnetization increases smoothly around $B_c$ in contrast to the abrupt enhancement of optical transitions. Hence, the electronic property of EuCd₂As₂ experiences sharp change while the Weyl band is continuously shifted by the external magnetic field.

determined by the mid-infrared (MIR) spectrum, the deduced model predicts three additional optical transitions and a crossing feature at higher energy, which are quantitatively verified by the near-infrared (NIR) spectrum. Our work provides a strategy for accessing and tuning the 3D VHS.

## Results

### Material realization

EuCd₂As₂ is chosen for the proposed physics due to the presence of strong exchange interaction[59] and magnetic Weyl semimetal phase[60–64]. Single crystal EuCd₂As₂ (a trigonal structure with space group of P3̄m1) is synthesized by using the standard flux method (more details in the "Methods" section). The structure and quality of EuCd₂As₂ crystals are confirmed by X-ray diffraction (XRD) and Raman spectroscopy (Supplementary Section I). As presented in Fig. 2a, the magnetization increases with the out-of-plane magnetic field until the saturation field $B_s$. At 2 K, $B_s$ is determined to be around 2 T. Both the saturation magnetization and saturation field agree with the previous experiments[65–67]. The magnetic ground state of EuCd₂As₂ is A-type antiferromagnetic (AFM) with Eu magnetic moments lying along the in-plane direction[66]. The magnetism of the EuCd₂As₂ originates from the large local Eu magnetic moments, while the transport property is dominated by the conduction electrons from the arsenic and cadmium orbitals[68]. The smooth magnetization variation with the external magnetic field in Fig. 2a suggests that the magnetic moments are continuously canted without sudden change. As depicted in Fig. 2b, EuCd₂As₂ enters the A-type AFM order below $T_N = 9.5$ K. When the magnetic moments align toward the $c$-axis, the spin degeneracy is lifted[62,69]. A pair of 3D Weyl nodes is generated from the crossings of two spin-polarized bands, as confirmed by the angle-resolved photoemission spectroscopy (ARPES)[34]. The relativistic carrier mass[59], nonlinear anomalous Hall effect[69,70], and topological Hall effect[71] are

observed. The topological phases in EuCd₂As₂ are tightly associated with the magnetic order[60,72,73]. Upon applying the out-of-plane magnetic field, the Weyl bands in EuCd₂As₂ are effectively shifted by the exchange interaction. The shifting energy increases with the canting of Eu magnetic moments and attains the maximum (on the order of 100 meV) at the magnetic saturation field[59]. Importantly, there exists only a single pair of Weyl nodes in the Brillouin zone in this magnetic configuration. Therefore, EuCd₂As₂ is suitable for realizing the proposed 3D VHS.

### Magneto-mid-infrared spectroscopy

VHS typically induces prominent optical features since the intensity of the optical transition and corresponding spectral peaks are strongly determined by the DOS. To study the band evolution and presence of the 3D VHS with the external magnetic field, we carry out the magneto-infrared spectroscopy on EuCd₂As₂ (refer to the "Methods" section). The magneto-infrared spectrum is obtained by a silicon bolometer in the National High Magnetic Field Laboratory (NHMFL, Tallahassee) with the out-of-plane magnetic field (Faraday geometry), as exhibited in Fig. 2c, where $R_B$ and $R_0$ are the reflectivity measured in magnetic field $B$ and the zero field, respectively. The spectrum shows two prominent optical features $T_\alpha$ and $T_\beta$ that systematically shift with the external magnetic field and reach saturation at high magnetic fields. However, these features are very weak at low fields. An abrupt enhancement of the optical features is observed in the critical field $B_c \approx 0.6$ T. This is unusual in magneto-infrared spectroscopy and calls for further validation to avoid possible artifacts from the experimental setup. Thus, we further perform magneto-infrared spectroscopy on a same-batch crystal in East China Normal University (ECNU, Shanghai) with entirely different setups from the detector (Mercury−Cadmium−Telluride, MCT) to the magnet (closed-cycle 12 T superconducting magnet). As exhibited in

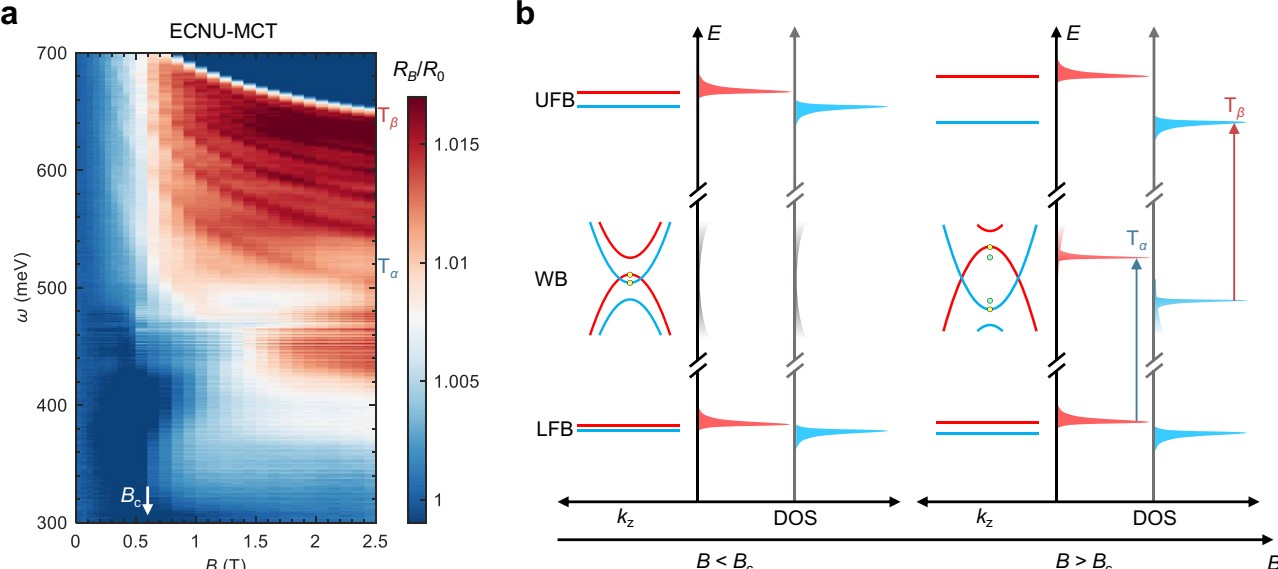

**Fig. 3 | Low-field magneto-infrared spectrum and schematic band structure of EuCd₂As₂. a** False-color plot of relative reflectivity $R_B/R_0$ of EuCd$_2$As$_2$ at low fields. **b** Band structure and DOS before and after the critical field $B_c$. Weyl bands (WB) are located at distant energy from the upper flat band (UFB) and the lower flat band (LFB). Red and blue distinguish the bands with opposite spins. For clarification, the DOS from each set of spin-polarized bands is plotted on the separate energy axis. Below the critical field, the saddle point at zero momentum (yellow dot) only

contributes to a weak spectral feature owing to the finite kink in DOS. Increasing external magnetic field leads to the larger exchange splitting and resultant higher Fermi velocity of the Weyl bands. As reaching the critical field $B_c$, the requirement of $v_z \geq \sqrt{2}v_{xy}$ is met in EuCd$_2$As$_2$ as proposed in the minimal model. The divergent DOS originates from the newly formed 3D VHS (green dot), leading to the abrupt enhancement of optical features T$_\alpha$ (blue arrow) and T$_\beta$ (red arrow), accompanied by satellite peaks from the inter-band Kerr effect.

Fig. 2d, all spectral features and their abrupt enhancement at $B_c$ are reproducible. Stacking plots of the raw spectra from both ECNU and NHMFL are provided in Supplementary Section IV. Meanwhile, satellite peaks become more distinguishable, covering a broad spectral range, but are still much weaker than T$_\beta$. Thus, we assign T$_\beta$ to the main feature due to its dominant spectral weight and the highest energy (refer to Supplementary Section V for more details). The apparent energy positions of T$_\alpha$ and T$_\beta$ are extracted from the spectrum measured in the above setups, as plotted in Fig. 2e and f, respectively. It is noteworthy that both the field-dependent energy variation of T$_\alpha$ and that of T$_\beta$ scale with the magnetization presented in Fig. 2a, which reveal the close relevance between electronic bands and magnetism. The field dependence of the optical transition energy (square in Fig. 2e, f) roughly overlaps with the linear scaling of magnetization (red and blue solid curves in Fig. 2e, f), similar to the previous report in EuTe[74]. The scaling parameters (placed near the corresponding curves) from the two setups are close to each other, indicating the quantitative reproducibility of the features. In the following, we mainly discuss the spectra measured in ECNU due to the better signal-to-noise ratio.

The experimental findings of EuCd$_2$As$_2$ are summarized as follows: (1) two optical features T$_\alpha$ and T$_\beta$ shift oppositely in energy accompanied by satellite peaks; (2) the energy of both transitions linearly scales with the out-of-plane magnetization rather than the external magnetic field; (3) their energy shift is as large as nearly 200 meV; (4) all spectral features undergo a "3-stage" intensity variation: keeping low below $B_c$, sharply increasing around $B_c$, saturating after the saturation field of magnetization $B_s$; (5) the magnetization increases continuously around $B_c$. Among these findings, (2) and (3) suggest that the band shift results from the exchange splitting and exclude the possible origin from the Zeeman splitting; (2), (4), and (5) indicate that the electromagnetic response of the system experiences an abrupt enhancement, although the energy band shifts continuously following the magnetization. These observations are in line with the VHS in the proposed model.

## Optical transitions assignment and emergence of 3D VHS
To further validate the abrupt behavior in EuCd$_2$As$_2$, measurements with higher field resolution were carried out at the low field regime (Fig. 3a). In Fig. 3a, the intensity of both T$_\alpha$ and T$_\beta$ increases sharply when $B \geq B_c$, which leads to a distinct boundary at $B_c$ (white arrows) in the false-color map. The main optical features are accompanied by a series of satellite peaks. Therefore, a strong increase in the overall spectrum is observed at $B_c$ in addition to the main features T$_\alpha$ and T$_\beta$. The Landau-level resonance is excluded for the origin of these satellite peaks since the energy of these peaks linearly scales with magnetization instead of the magnetic field. Subtle optical transitions can be found below the critical field, but the intensity of these features increases sharply near the critical field. The large transition energy indicates that the optical transitions do not originate solely within the energy range of the Weyl bands since the energies of optical transitions rising from Weyl bands are typically within 100 meV[56,75–77]. According to previous density functional theory (DFT) calculations[34,62,69], several energy bands possessing local flat dispersion exist at much higher and lower energy ranges (with respect to the Weyl bands) within the momentum range of the Weyl nodes. These bands have the potential to act as the initial or final states for the observed optical transitions with energies up to several hundreds of meV. Here, the local flat band only requires flat dispersion around Γ point, which differs from the complete flat bands throughout the Brillouin zone.

The multiple flat bands from the 4f orbital of Eu atoms observed in ARPES[34,59] might not contribute to the observed transitions due to the discrepancy in energy (>1 eV) and linewidth (~1 eV). As described in more detail in Supplementary Section VII, the opposite trends of T$_\alpha$ and T$_\beta$ as well as the distant energy positions of their zero-field extrapolation, suggest the presence of both the upper flat band (UFB) and the lower flat band (LFB). Thus, we plot both flat bands and Weyl bands in the schematic drawing of Fig. 3b. Due to the exchange interaction, both the UFB and LFB are expected to experience exchange splitting (energy bands with opposite spins are distinguished in blue and red), similar to the magnetic Weyl bands[69].

At a small magnetic field (left panel of Fig. 3b), the Weyl bands possess the saddle points at zero momentum with non-divergent DOS. Therefore, the DOS is continuous and low throughout the effective energy range of Weyl nodes. The experimental spectrum exhibits negligible variation. It is noteworthy that $T_\alpha$ and $T_\beta$ can be traced below the critical field and undergo the "3-stage" intensity variation since the critical point at zero momentum can also provide a finite kink in the DOS. By further ramping up the magnetic field, the Fermi velocity $v_z$ increases, as discussed. Due to the large exchange splitting, the band shift reaches the order of hundreds of meV within the experimental field range, which results in a prominent increase in the Fermi velocity $v_z$ while keeping $v_{xy}$ constant. Once $B \geq B_c (v_z \geq \sqrt{2} v_{xy})$, the original saddle point becomes a band extremum (yellow dot), while a new saddle point (green dot) appears. These two critical points act as Lifshitz points with different topological properties, as shown in Fig. 1. When the Fermi level shifts through either one, the topology of the Fermi surface varies (Lifshitz transition). Especially, the Fermi surface undergoes a topological Lifshitz transition[58] with its Chern number changing from 0 to $\pm 1$, only if the Fermi level drops through the newly formed saddle point, although the VHS is not topologically protected. More details regarding the topological Lifshitz transition are provided in Supplementary Section XIII. The saddle point is located at finite in-plane momentum and serves as a 3D VHS because of the effective dimension reduction along the tangential direction. It allows the prominent optical transitions between the 3D VHS and the flat bands at distant energy positions because the optical transitions are sensitive to the divergent DOS even if they are far away from the Fermi level and not detectable in transport. As a result, $T_\alpha$ and $T_\beta$ exhibits a "3-stage" intensity evolution, including a significant intensity amplification around the critical field $B_c$. The minimal model successfully further reproduced the field-dependent intensity, as shown in Supplementary Section X.

The main optical features are accompanied by a series of satellite peaks which are especially noticeable for $T_\beta$. The presence of the densely distributed satellite peaks leads to a generally sharp increase in the reflectivity intensity of the overall spectrum at $B_c$, not only near the energy positions of $T_\alpha$ and $T_\beta$. We attribute the satellite peaks to the presence of Kerr rotation for the following reasons. On the one hand, the significant exchange splitting of bands in EuCd$_2$As$_2$ could lead to the different reflectivity between left- and right-handed circularly

polarized lights, which is the physical origin of the Kerr rotation. On the other hand, the spectral response of the Kerr rotation depends on the wavelength and the magnetization. A direct consequence is the presence of satellite peaks. As a smoking-gun characteristic, their energy separations between the adjacent satellite peaks should reduce as energy increases until satellite peaks merge with the main features corresponding to the inter-band transitions[78,79]. The energy evolution and magnetization-dependence of $T_\beta$ and satellite peaks agree with the Kerr rotation. More analysis of the satellite peaks is provided in Supplementary Section V.

When the external magnetic field further increases, the spin-up bands (red) continue to shift along the energy axis while the spin-down bands (blue) shift oppositely, as displayed in Fig. 3b. For the Weyl bands in our experimental configuration (Faraday geometry), we firstly focus on the spin-conserved optical transitions. Hence, Pauli-blocking allows two specific optical transitions. Without loss of generality, $T_\alpha$ is assigned as the transition between the spin-up LFB and upper VHS (blue arrow). $T_\beta$ corresponds to the transition between the lower VHS and spin-down UFB (red arrow). It should be noted that the proposed model is fully particle-hole switchable, meaning that the optical response is identical when the energy positions of all considered bands are inverted. We plot Fig. 3b, taking into account the consistency with ARPES results[80]. Due to distinct exchange splitting strengths, the energy bands from different origins generally shift with the magnetic field by different energies. The energy of $T_\alpha$ increases with the magnetic field, which implies a significantly larger energy shift of the VHS compared to the LFB. Similarly, the opposite trend of $T_\beta$ suggests the much larger energy shift of the UFB compared to the VHS. With the observed $T_\alpha$ and $T_\beta$ assigned as illustrated in the right panel in Fig. 3b, both the sharp variation at $B_c$ and energy evolution of the inter-band transitions in the MIR spectrum can be qualitatively understood. The prominent peak height enhancement around the critical field agrees well with the theoretical result of the abrupt DOS increase induced by the formation of 3D VHS (Supplementary Section X).

## Model prediction and spectroscopic verification

To better verify the model, we extract the band parameters based on the MIR experimental results, with which new optical features are predicted at a quantitative level. As schematically shown in Fig. 4a, besides the observed $T_\alpha$ and $T_\beta$, more optical transitions are predicted.

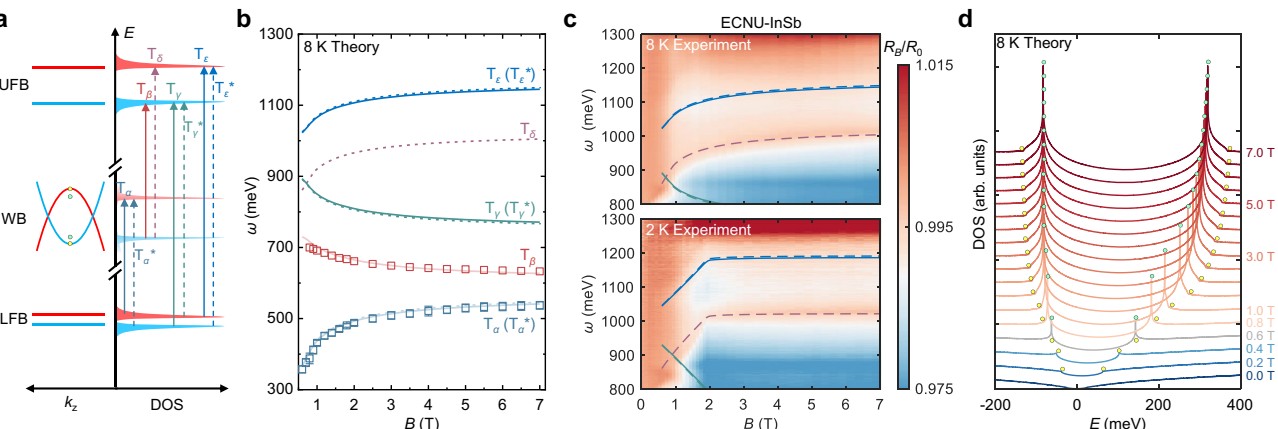

**Fig. 4 | Theoretical prediction of near-infrared optical transitions based on the deduced model and experimental verification. a** The prediction of additional optical transitions based on the deduced band structure. Solid and dashed arrows denote spin-conserved and spin-flipped optical transitions, respectively. **b** The predicted photon energy and magnetic field evolution of optical transitions at 8 K. The model parameters are experimentally determined by the MIR spectrum. Three additional optical transitions $T_\gamma, T_\delta, T_\varepsilon$ and spectral crossing around 1 T are predicted. Solid and dashed curves correspond to the transitions in (**a**). The squares

denote the energy of observed transitions in the MIR regime. **c** Magneto-NIR spectrum measured at different temperatures. The experimental result agrees with the model prediction (solid and dashed curves). The agreement is better verified in 2 K data because of the sharper flattening of the magnetization around the saturation field. **d** Calculated DOS of the Weyl bands in EuCd$_2$As$_2$ based on the extracted band parameters. A pair of 3D VHS (green dot) is formed above $B_c$, leading to the abrupt enhancement of inter-band transition.

The optical transitions between those flat bands might present spectral features in the high energy spectrum as denoted by green and deep blue arrows labeled $T_\gamma$ and $T_\varepsilon$, respectively. When the spin mixing effect at finite momentum is taken into further consideration, the optical transitions are allowed to occur between the two states with opposite spins, which leads to the splitting of each transition[57]. These spin-flipped transitions are denoted by the dashed arrows. The energy splitting of $T_\alpha, T_\gamma, T_\varepsilon$ should be identical and much weaker compared with that of $T_\beta$. Although $T_\alpha$ is located at the center of the measurable spectral range at the saturation field, no sign of energy splitting is observed, which suggests the negligible exchange splitting and the energy shift of the LFB. It agrees with the opposite energy evolution between $T_\alpha$ and $T_\beta$, from which the energy shift of the LFB is deduced to be the smallest among all associated bands. It also explains why the Kerr-effect-induced satellite peaks are not resolved for $T_\alpha$[81,82]. Therefore, the splitting of $T_\alpha, T_\gamma, T_\varepsilon$ are not expected to be found in the spectrum regardless of the spin configuration (therefore, the spin-flipped transitions are labeled as $T_\alpha^*, T_\gamma^*, T_\varepsilon^*$). On the contrary, the splitting of the UFB and resultant $T_\beta$ are expected to be significant. Thus, the model predicts $T_\delta$ as a spectral counterpart of $T_\beta$, which stays at a much higher energy position and evolves oppositely with the magnetic field. Although the finite spin mixing effect weakens the oscillator strength of the spin-conserved optical transitions, the influence of divergent DOS dominates the spectral variation. As a result, the intensity of all transitions monotonically rises with the magnetization.

Figure 4b shows the prediction of optical transitions with the following features: (1) three additional optical features are expected to be detected in the NIR regime, originating from $T_\gamma, T_\delta, T_\varepsilon$; (2) the transition energy of $T_\delta$ and $T_\varepsilon$ should increase with the magnetic field, while that of $T_\gamma$ behaves oppositely. All new transitions should scale with magnetization, similar to $T_\alpha$ and $T_\beta$; (3) spin-flipped optical transitions $T_\alpha^*, T_\gamma^*, T_\varepsilon^*$ should merge with their spin-conserved counterparts ($T_\alpha, T_\gamma, T_\varepsilon$) due to the negligible splitting of the LFB; and (4) the transition energy of $T_\gamma$ and that of $T_\delta$ cross with each other around 1 T.

The prediction is made based on the band parameters determined by: (1) the zero-field energy difference between the Weyl band extrema and the LFB/UFB estimated by the extrapolation of $T_\alpha$ and $T_\beta$, (2) the negligible splitting of the LFB as discussed, (3) the value of band parameter $\Delta$ at the zero field given by the previous report[34] of ARPES and scanning tunneling microscopy (STM) study, and (4) field-dependent exchange splitting and other band parameters $m$, $v_{xy}$ derived from the magnetization data, the energy variation of $T_\alpha$ and $T_\beta$ as well as the critical field $B_c$ in the experiment. The values of all extracted band parameters are given in Supplementary Section VIII.

To verify the model prediction quantitatively, we perform NIR magneto-infrared spectroscopy in ECNU. The top panel of Fig. 4c displays the false-color map of the NIR spectrum measured at 8 K. The solid and dashed curves denote the theoretical prediction from Fig. 4b. All predicted optical transitions $T_\gamma, T_\delta, T_\varepsilon$ are observed around the expected energy positions, especially the most prominent $T_\delta$. The crossing between $T_\gamma$ and $T_\delta$ is verified near the expected magnetic field. To overcome the comparatively weak spectral intensity of $T_\gamma, T_\varepsilon$ and better verify the model prediction, we examine the spectrum at a lower temperature due to the sharper change in the magnetization behavior around the magnetic saturation field. As shown in the bottom panel of Fig. 4c, the predicted optical features at 2 K are better identified. The raw spectra and zoom-in false-color plots in the specific energy range are provided in Supplementary Section XI. During the prediction of the 2 K results, we fix most of the band parameters but change the exchange splitting parameter based on the 2 K magnetization data depicted in Fig. 2a. Notably, the optical transitions between two Weyl bands are not resolved in the spectrum which might be forbidden owing to the combination of parity and spin configuration of the associated two Weyl bands which originate from $s$- and $p$-orbitals with opposite spins. Meanwhile, the VHS-irrelevant optical transitions

($T_\gamma, T_\varepsilon$) present a much lower spectral weight compared with the VHS-originated transitions. It is most likely because both the UFB and LFB are not ideally flat, which results in the broadening and the decline of the peak in DOS. In contrast, the VHS serves divergent DOS, leading to comparatively stronger spectral intensity of $T_\alpha, T_\beta, T_\delta$.

On the basis of the validated model with experimentally extracted band parameters, we theoretically calculate the DOS within the Weyl bands at different magnetic fields, as plotted in Fig. 4d. The DOS is calculated in 0.01 meV energy resolution based on the 3D momentum mesh grid with $2 \times 10^{-4}$ nm$^{-1}$ interval. A pair of VHS (green dot) appears at the critical field (gray curve at 0.6 T), then shifts oppositely in energy with enhanced intensity and eventually approaches saturation at a high field. Notably, the VHS develops only above the critical field, while the critical point at zero momentum (yellow dot) exists throughout the whole magnetic field range. The latter only leads to a kink in DOS, similar to that in most 3D systems. In the spectrum with a better signal-to-noise level, the optical contribution from such a kink is resolved, which explains the presence of weak spectral characteristics below $B_c$ followed by a sharp increase in intensity after that.

## Discussion

Generally, for the quadratic energy bands in 3D, the VHS is thought to be absent, which is determined by the state distribution on $|\mathbf{k}|$ in 3D momentum space[1]. In principle, VHS might be realized by tuning the band dispersion to the quartic or even higher order along all three directions, and it requires simultaneously being the extreme point along all directions following the definition of VHS[1] ($|\nabla E(\mathbf{k})|_{\mathbf{k}_{VHS}} = 0$). The high-order dispersion is accessible in 2D (e.g. few-layer graphene[83]) but remains more challenging in 3D. Instead, we turn to focus on the external field tunability of magnetic Weyl semimetal and achieve 3D VHS by effectively reducing the quasi-particle dimension. The EuCd$_2$As$_2$ system possesses similar dispersion along $k_x$ and $k_y$[59]. The in-plane isotropy is crucial for accessing the 3D VHS (see Supplementary Section III for the phase diagram and predicted spectra in anisotropic cases). It turns the in-plane dispersion into the Mexican hat shape above the critical field. As a result, a loop of saddle points forms along the in-plane direction, the so-called saddle ring. Most importantly, the extreme point of the Mexican hat is also the critical point of the third dimension. In the vicinity of this 3D VHS, we examine the dispersion along all three directions. The band structure disperses oppositely along $k_z$ and the in-plane radial direction. However, it is flat along the in-plane tangential direction because of the in-plane isotropy. As a result, the dimension along the tangential direction is effectively eliminated, and an effective 2D saddle point emerges along the $k_z$-radial plane. It causes logarithmically divergent VHS to appear in EuCd$_2$As$_2$ with the strict 3D band structure that is distinguished from the quasi-2D system. More details on the dispersion of 3D VHS are provided in Supplementary Section III. The realized 3D VHS does not fall into any current category of VHS but becomes P$_1$-type-like stemming from the effective dimension reduction (refer to Supplementary Section II for the classification and notation of VHS in different dimensions). In a more extreme case, the 3D VHS with 1D-like divergent DOS (Q$_0$- and Q$_1$-type) might be realized in a 3D isotropic band with Mexican-hat dispersion since the dispersion around the VHS exhibits flat behavior along the two dimensions. More details on the classification of VHS are provided in Supplementary Sections II and III. Meanwhile, it needs to be emphasized that the observed optical transitions are associated with the higher-energy saddle points of the Weyl bands instead of the low-energy Weyl nodes. Although the minimal model of Weyl semimetal is adopted, the presence of Weyl nodes is not the prerequisite for accessing 3D VHS. Instead, the (nearly) isotropic saddle ring is necessary.

We noticed controversies regarding the metallicity/insulativity of the EuCd$_2$As$_2$ system. While the Weyl nodes are directly observed in ARPES[34] and semimetallic transport phenomena[59,70,71] are reported,

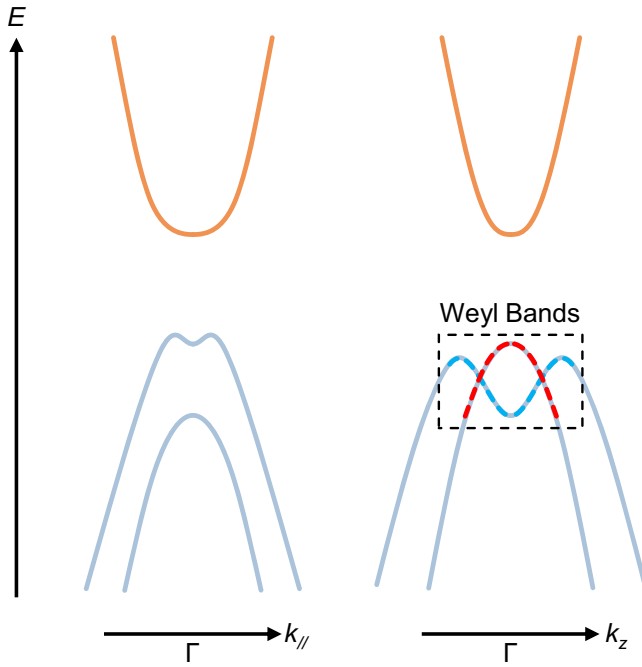

**Fig. 5 | More realistic band structure around Γ point.** The left and right panels are schematic plots of band structure along the in-plane $k_\parallel$ and out-of-plane $k_z$ directions, respectively. Orange (azure) curves denote the conduction (valence) bands. The proposed model might resolve controversies regarding the metallicity/insulativity of the EuCd$_2$As$_2$ system and explain the discrepancy in previous ARPES, transport, and optical results. The Weyl bands within the black dashed box are associated with the features in magneto-infrared spectroscopy. Red and blue dashed curves denote the band with opposite spin textures.

recent spectroscopy and transport research present the semiconducting nature of EuCd$_2$As$_2$[80,84] and point out the sensitivity of transport characteristics upon shifting the chemical potential. However, the Weyl dispersion should not be simply driven into the trivial one by tuning the chemical potential. Nevertheless, our experimental findings and physical model are not influenced by this controversy due to the high-energy nature of the observed transitions and the insensitivity of the model to the chemical potential.

To account for these observations from different techniques and resultant controversy, we extend the minimal model to a more realistic band structure shown in Fig. 5. At a small energy range within the top of the valence band, 3D Weyl nodes appear along $k_z$ direction. For $k_z = 0$, the dispersion along $k_\parallel$ direction should not exhibit a crossing structure. At a large energy scale, all related bands bend downwards, forming a band structure of the so-called "Weyl semiconductor"[85,86]. The realistic version of the model might help to resolve the controversy. The Weyl nodes along $k_z$ direction in the realistic version of the model explain (i) the observation of Weyl nodes along $\Gamma - A$[34]; (ii) the absence of the Weyl nodes along $k_\parallel$ direction[34,80]; (iii) Weyl-related transport phenomena[59,70,71]; (iv) metallic infrared response[65] (v) the observation of VHS in our spectrum. On the other hand, the gapped structure in the model explains (i) the gap observed in the result of time-resolved APRES[80]; (ii) gapped features in infrared/transport experiments[80,84,87,88]; (iii) large-energy optical transitions in our spectrum. Based on the model, it is natural to expect sensitivity of transport property upon shifting the Fermi energy. In comparison, high-energy optical features should not be sensitive to the Fermi energy. In the EuCd$_2$As$_2$ system, the Fermi energy can be shifted by different growth methods[34,84], condition of impurity level and self-doping effect[80], surface absorption effect[89]. With the proposed band structure, the insulating behavior can be understood in the light hole-doping case.

While the hole-doping level increases, the system undergoes an insulator-to-metal transition, and Weyl nodes can contribute to the transport properties, supporting the reported Weyl-related transport phenomena[59,70,71]. As for our experimental observations, the underlying physics is the evolution of Weyl bands (dashed box in Fig. 5) controlled by the exchange interaction, independent of the Fermi energy variation. The band edge of the conduction band probably plays the role of UFB and acts as the final state for transition T$_\beta$.

For the previously reported systems with VHS, much effort has been devoted to shifting the Fermi level position by applying an electric field or chemical doping[15,18,90]. Meanwhile, it remains a substantial challenge to tune the VHS itself. The main strategy is to synthesize different crystals and search for new materials possessing VHS naturally close to the Fermi level. The twisted van der Waals material acts as limited cases to achieve controllable VHS[14,15,91] by twist angle and displacement field. However, it requires sophisticated twisted and double-gating devices. Electric gating becomes less efficient in 3D systems due to the strong screening. Here, the discovered VHS in EuCd$_2$As$_2$ exhibits high and in-situ tunability by the external field. The modification of the Weyl bands (on the order of 100 meV) driven by the spin-canting-induced exchange splitting serves as the physical origin of this tunable VHS. This approach outperforms the Zeeman-effect-driven magnetic Weyl semimetals owing to the much higher energy of band shift.

In addition to the EuCd$_2$As$_2$ system, the proposed model seems to be general and applicable to a variety of material systems[92]. However, it might be difficult to distinguish such VHS from other optical contributions in those systems with fixed band parameters. In our approach, the special tunability of the magnetic Weyl system enables the magnetic field to control VHS, especially its presence and absence, thus excluding many other possible mechanisms. For example, the field-induced gap variation should not exhibit sudden enhancement of transition intensity at the critical field. If the energy dispersion or the magnetism can be influenced by the growth conditions or chemical doping, the critical field might be shifted or eliminated in EuCd$_2$As$_2$. As for the correlation transport property, it is not expected in EuCd$_2$As$_2$ because the Fermi level is away from the VHS. Further investigation by the STM under magnetic fields might serve as an alternative probe for the 3D VHS in EuCd$_2$As$_2$. Based on the tunability of the described strategy, 3D VHS can be tuned close to the Fermi energy in other systems, which might help to establish overlooked 3D systems for studying strong correlation physics in the future. After submitting this work, we became aware of similar magneto-infrared data about EuCd$_2$As$_2$[80].

## Conclusion

In summary, we report the formation of 3D VHS in the topological magnet EuCd$_2$As$_2$ with spin canting. While the Weyl bands are continuously shifted by the external magnetic field due to the exchange interaction between the itinerant electrons and local magnetic moments, the Fermi velocity increases and reaches the critical value. The emergence of 3D VHS is evidenced by the abrupt intensity increase of optical transitions in the MIR magneto-infrared spectrum. The overall spectral features are quantitatively explained by the two-band minimal Weyl model. Based on the band parameters determined in MIR experiments, the model predicts three additional features and a spectral crossing which are fully verified by the NIR magneto-infrared spectroscopy. Our result provides a strategy for realizing VHS in 3D systems and accessing the tunability of 3D VHS by the external field.

## Methods

### Single-crystal growth

The EuCd$_2$As$_2$ single crystals were synthesized by using the Sn-flux method. Starting materials Eu, Cd, As, and Sn (1:2:2:10 in molar ratio) were placed into an alumina crucible. The crucible with the filter

(quartz wool) was sealed in a vacuum quartz ampoule and then heated to 900 °C in 12 h followed by holding for 24 h to reach sufficient reaction. After the ampoule slowly cooled to 500 °C at a rate of 2 °C/h, liquated Sn was removed by a centrifuge. The millimeter shiny crystals were selected from quartz wool.

## Crystal characterization

The crystal structure was examined by the XRD (PANalytical Empyrean). The Raman-active phonon modes were detected with 632.8 nm He–Ne laser excitation in a home-built Raman spectroscopy system at room temperature. The magnetization measurement was performed in a magnetic property measurement system (MPMS, Quantum Design).

## Magneto-infrared measurement at NHMFL

Magneto-infrared spectroscopy was performed at NHMFL by a Fourier-transform infrared (FTIR) spectrometer (Bruker 80 V) and a 17.5 T liquid-helium-cooled superconducting magnet. The collimated MIR beam from a globar light source was guided through a brass light pipe in a vacuum and focused on the (0 0 1) surface of the $EuCd_2As_2$ crystal with the out-of-plane DC magnetic fields (Faraday geometry). The reflected infrared beam was collected by the nearby bolometer, and then spectra were obtained by Fourier analysis.

## Magneto-infrared measurement at ECNU

Magneto-infrared spectroscopy was performed at ECNU by the FTIR spectrometer (Bruker 80 V) and a 12 T closed-cycle superconducting magnet (Oxford Instruments TeslatronPT). The collimated MIR (NIR) beam from a globar (tungsten) light source was guided into the variable temperature insert (VTI) of the magnet and then focused on the (0 0 1) surface of the $EuCd_2As_2$ crystal by an on-axis parabolic mirror. The experiment configuration is identical to that in NHMFL. The reflected infrared beam was propagating out of the VTI with the guidance of a brass light pipe and collected by the liquid-nitrogen-cooled detector located near the magnet. MCT and InSb detectors are used for MIR and NIR spectra, respectively.

## Minimal model and DOS calculation

Introducing $H_{exc} = \bar{g}B_{exc}\sigma_z + \delta gB_{exc}\sigma_0$ that describes exchange interaction[52] between the itinerant electron and local magnetic moment, the overall Hamiltonian reads

$$H = H_0 + H_{exc} = \begin{pmatrix} \Delta - m\mathbf{k}^2 & v_{xy}(k_x - ik_y) \\ v_{xy}(k_x + ik_y) & -\Delta + m\mathbf{k}^2 \end{pmatrix} + \begin{pmatrix} \delta gB_{exc} + \bar{g}B_{exc} & 0 \\ 0 & \delta gB_{exc} - \bar{g}B_{exc} \end{pmatrix} \tag{2}$$

where $\sigma_{x,y,z}$ represent the Pauli matrices with the unit matrix $\sigma_0$; $\Delta$, $m$, $v_{xy}$ are the band parameters for the given materials with $\hbar = 1$; the 3D vector $\mathbf{k}$ refers to the momentum. $B_{exc}$ is the exchange field proportional to the magnetization under the mean-field approximation. $\bar{g}$ and $\delta g$ are the average and difference of the effective $g$ factor given by the two Weyl bands from the different orbitals. The energy dispersion tuned by the exchange field is given by

$$E(\mathbf{k}) = \delta gB_{exc} \pm \sqrt{\left(\Delta + \bar{g}B_{exc} - m\mathbf{k}^2\right)^2 + v_{xy}^2\left(k_x^2 + k_y^2\right)}. \tag{3}$$

The Fermi velocity of the Weyl node along $k_z$ direction becomes tunable following $v_z = 2\sqrt{(\Delta + \bar{g}B_{exc}) \cdot m}$, because introducing an exchange interaction term $\bar{g}B_{exc}\sigma_z$ is equivalent to modifying the band parameter $\Delta$. With all band parameters determined from the magneto-infrared spectrum, the magnetic-field-dependent DOS spectrum is

calculated numerically. Momentum states are set up by a 3D mesh grid with $2 \times 10^{-4}$ nm$^{-1}$ interval. With the corresponding energy obtained by the energy dispersion in Eq. (3), the DOS is given by the statistical distribution of momentum states with an energy resolution of 0.01 meV. The artificial random spikes are found in the resultant DOS spectrum because of the limitation of numerical calculation but are no longer resolved after the adjacent-average smoothing with a 5-point span (0.04 meV energy span). More details of the model are given in Supplementary Sections VI and XII.

## Data availability

The data generated in this study are provided in the Source Data file. All other data that support the findings of this study are available from the corresponding author upon request. Source data are provided with this paper.

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

## Acknowledgements

We thank Hugen Yan, Yaoming Dai, Jianhui Zhou, Chun-Gang Duan, Yanmeng Shi, Ryosuke Akashi, Changyoung Kim, Wei Ku, Qian Li, Hua Jiang, Huilin Lai, and Wei Lu for their helpful discussions. X.Y. was sponsored by the National Key R&D Program of China (Grant No. 2023YFA1407500) and the National Natural Science Foundation of China (Grant Nos. 12174104 and 62005079). C.Z. was sponsored by the National Key R&D Program of China (Grant No. 2022YFA1405700), the National Natural Science Foundation of China (Grant Nos. 12174069 and 92365104), and the Shuguang Program from the Shanghai Education Development Foundation. Z.Y. was sponsored by the National Natural Science Foundation of China (Grant No. 12174455) and the Natural Science Foundation of Guangdong Province (Grant No. 2021B1515020026). Y.H. was sponsored by the National Natural Sciences Foundation of China (Grant Nos. 12104518 and 92165204) and GBABRF-2022A1515012643. A portion of this work was performed at the National High Magnetic Field Laboratory, which is supported by the National Science Foundation Cooperative Agreement No. DMR-1644779 and the State of Florida.

## Author contributions

X.Y. conceived the idea and supervised the overall research. Y.D., X.M., X.J., G.W., C.H. and X.W. synthesized $EuCd_2As_2$ single crystal and performed characterization. Y.W., W.W. and C.Z. conducted magnetization and temperature-dependent resistivity measurements. M.O. and W.W. carried out the magneto-infrared experiments in NHMFL. Z.S. set up the magneto-infrared system in ECNU. W.W., P.Z., Z.S., J.C., C.P. and H.P. participated in the construction of the magneto-infrared system. W.W. and Z.S. measured magneto-infrared spectra in ECNU. W.W. and X.Y. analyzed magneto-infrared spectra based on the minimal model with the help of Z.Y., X.-S.N., Y.H., H.-Z.L. and R.Y. W.W., X.Y., C.Z. and Y.X. wrote the paper with all coauthors' help.

## Competing interests

The authors declare no competing interests.
