## [Peer Review File · Nature Communications]

The discovery of three-dimensional Van Hove singularityReviewer #2 (Remarks to the Author):

This manuscript presents the magneto-infrared spectroscopy results revealing the Van Hove singularity (VHS) in bands of a three-dimensional EuCd₂As₂ under a magnetic field. The authors claim that the VHS appears at magnetic fields greater than 0.6T. The data quality seems to be high, and overall discussions are well written. The concept that the VHS is tuned by magnetic field for the Weyl state bands is very intriguing. If the claim by the author is correct, the results of the discovery of three-dimensional Van Hove singularity would be suitable for publication in Nature Communications. However, I cannot decide whether the current manuscript is suitable for publication or not because of the not-enough presentation. Several concerns described below should be addressed before the judgment.

- The previous ARPES and DFT calculations [34] demonstrated that the flat bands of EuCd₂As₂ are located deeper than 1eV in binding energy. It is incompatible with the claim by the current authors that it is about 700meV according to the T_{beta} data. The authors should discuss this contradiction.

- The authors assume very simple flat bands (one each for the occupied and unoccupied sides in the case without a magnetic field). However, ARPES and DFT calculations demonstrate many of those due to multiple Eu4f levels. Furthermore, these 4f bands split over a wide energy range of about 1eV. The authors should point out this fact in the text and explain why these multiple Eu4f levels should not be considered.

- The authors attribute the satellite peaks to the Kerr effects but do not explain why they believe so. The data quality of Fig.2d looks high enough to extract the detailed peak positions and their evolution with an external magnetic field. I request that the authors show not only color maps but also spectra, and explain the energy dispersion, their separation, and how to understand those based on the Kerr effects in the main text.

- I can understand that the peak positions of T_{beta} can be defined precisely from the data in Fig.2d but I wonder if it is possible for T_{alpha}. More seriously, I suspect that T_{alpha} and T_{beta} could not be determined from Fig2c with poor quality. In particular, the error bar is not indicated (or negligible) in Fig.2e. These suspects come from the not-enough presentation of the authors plotting only color images, not spectra. I request that the authors show spectra, not only images (Fig.2c,2d), and demonstrate how they determined T_{alpha} and T_{beta}.

- Related to the above, it was unclear why the authors assigned the highest energy of T_{beta} as the main peak and the others as satellites. As mentioned above, Eu4f levels are possibly the reason for these many peaks.

- The ECNU data are of much higher quality than the NHMFL data. The only advantage of the ECNU data is to double-check their conclusion. In that case, I request that the authors move the NHMFL data to supplemental information, and comment only briefly in the main text that the conclusion is reproduced by another data set.

- The authors claim that T_{delta} and T_{gamma} cross but it is unclear to me. A better presentation using real spectra (not only color plots Fig4c) is necessary to convince the readers. I request that the authors plot spectra, not only color images, and justify this claim.

- Related to the above, T_{gamma}, T_{delta}, and T_{epsilon} are all unclear to me. To justify their claim, the author should display spectra not only color images for Fig4c, and justify their claims that these are indeed obtained from the data.

In conclusion, I will reconsider whether the data are sufficient to meet the authors' claims after the major revision.

Reviewer #3 (Remarks to the Author):

Wu et al. present a magneto-optical study on EuCd₂As₂ up to 12 T, primarily at low-temperature AFM phase. The authors proposed a model for their observations based on the appearance of a 3D VHS in this material. The request for this model is that the system is a Weyl semimetal and the nodes can be tuned by the external magnetic field. The Weyl semimetal nature of this compound is widely accepted, especially based on ARPES measurements that probe the band structure and report Dirac nodes in the vicinity of the Fermi energy. However, one should point out that these measurements are more surface-sensitive.

The idea of 3D VHS in a real system is exciting, especially in a system that can be easily tunable with an external magnetic field. However, to my opinion, here authors try to explain their results with biased knowledge and at certain points fail to provide a full picture of the observations. Therefore, I cannot recommend this work for publication in Nature Communications without a more thorough examination of the presented data. Here are some of my concerns that the authors should consider:

- Authors claim that there is a critical magnetic field of ~ 0.6 T initiates certain optical transitions that are in principle at the heart of the discussions. However, a quick inspection of Fig 2 a and b reveals that the T β transitions in fact extend all the way to the zero field (with MCT detector, obviously bolometer was not sensitive enough), whereas the T α transitions cannot be traced because the lowest energy that the authors presented is 300 meV that is quite large in principle. Therefore, it is not clear if the features evolve gradually or appear at so called B_c.
- Authors never show a raw spectra that would allow one to assess the spectra by itself in terms of signal-to-noise ratio. I think it is important for the previous point, as well.
- Authors discuss basically the optical transitions between saddle points arise due to the band crossings and gradual move of these points with magnetic field. Looking to the band structure of EuCd₂As₂ one would expect these crossings at the very low energy region. However, authors present transitions located at quite a large energy regime. How these are directly related with the low energy features of the Weyl nodes?
- Authors does not provide any resistivity curves for their sample, but considering that is available in the literature one would assume that they have a seemingly metallic compound. In this case how the cyclotron motion of the free carriers are coming to the discussion? Could it be that T α is in fact this cyclotron frequency rather than some optical transition?
- Finally, I would like to make a general remark on this compound. Although the Weyl semimetal picture seems to be widely accepted widely accepted situation for EuCd₂As₂, it is recently challenged with the optical studies itself (ArXiv:2301.08014). In fact, these are magneto-optical studies up to 16 T and directly related to this study, which is never mentioned by the authors in the manuscript in any context.

I should separately emphasize that the work at the last point is not peer reviewed and therefore I don't intend to make the authors accountable for these results, but it is in fact very important work because it directly provides an observation of the bulk properties of the material and gives evidence that the impurity effects are extremely important for this compound. Whereas the self doping effects could render the resistivity metallic, overall bulk material is a semiconductor with the band gap of on the order of ~ 700 meV. With magnetic field it is reduced but never closes suggesting that the band crossings creating Weyl nodes do not exist.

In fact, this interpretation of the spectra do not contradict with an earlier infrared spectroscopy data (PRB 94, 045112 (2016)), where a metallic sample is used similar to the common literature (presumably also similar to the one in this work). Here a very small contribution of the Drude component along with the high energy absorption with an onset at high energy regime can very well be interpreted as a slightly doped semiconducting behavior. Overall, I believe the current compound is not very trivial and authors should take into account the possible impurity and self doping effect of the compound.

REVIEWER COMMENTS

Reviewer #1 (Remarks to the Author):

In the manuscript entitled “The discovery of three-dimensional Van Hove singularity”, the authors report the discovery of 3D VHS in a topological magnet EuCd_2As_2 under the tuning of an out-of-plane field. By magneto-infrared spectroscopy, emergent optical features are found at a critical field, which are recognized by the authors as a Lifshitz transition caused by VHS. The manuscript is well written, and the physics is interesting. I recommend the publication of this manuscript in Nature Communications. But before that, I think the authors should answer my concerns raised below.

1. According to Eq. (1) and Fig. 1, I think the 3D VHS in DOS is NOT due to any critical point as claimed by the authors. Instead, it should be caused by a critical line (or more specifically *critical ring* for this work). Critical points in energy dispersion have been systematically studied [e.g. Nat Commun. 10, 5769 (2019), Phys. Rev. B 101, 125120 (2020)], while I do not think Eq. (1) at $v_z \geq \sqrt{2}v_{xy}$ belongs to any type of them.

Based on the arguments above, I also have the following questions.

2. What is the role of topology in this 3D VHS? Will it protect the 3D VHS?

3. In fact my answer to question 2 is no. I realize that the in-plane isotropy is crucial for Eq. (1) to produce a critical ring in energy dispersion. If, for example, we apply pressure along one in-plane direction only to break the in-plane isotropy, what will we observe in experiments?

Response to Reviewers' Comments

Reviewer 1

General comments:

In the manuscript entitled “The discovery of three-dimensional Van Hove singularity”, the authors report the discovery of 3D VHS in a topological magnet EuCd2As2 under the tuning of an out-of-plane field. By magneto-infrared spectroscopy, emergent optical features are found at a critical field, which are recognized by the authors as a Lifshitz transition caused by VHS. The manuscript is well written, and the physics is interesting. I recommend the publication of this manuscript in Nature Communications. But before that, I think the authors should answer my concerns raised below.

Response:

We sincerely appreciate the reviewer for the encouraging comments and insightful suggestions. We have thoroughly studied your suggestions and tried our best to revise the manuscript. In the revised version, we have improved the manuscript in the following aspects:

- 1) Analyzing the type of 3D VHS following the framework provided by *Nat Commun. 10, 5769 (2019)*.
- 2) Discussing the correlation between 3D VHS and topological Lifshitz transition.
- 3) Analyzing the band structure with in-plane anisotropy and predicting corresponding experiment observations.

Comment 1:

*1. According to Eq. (1) and Fig. 1, I think the 3D VHS in DOS is NOT due to any critical point as claimed by the authors. Instead, it should be caused by a critical line (or more specifically critical ring for this work). Critical points in energy dispersion have been systematically studied [e.g. *Nat Commun. 10, 5769 (2019)*, *Phys. Rev. B 101, 125120 (2020)*], while I do not think Eq. (1) at $v_z \geq \sqrt{2}v_{xy}$ belongs to any type of them.*

Based on the arguments above, I also have the following questions.

Response:

We fully agree with the reviewer on the origin of 3D VHS. The studied 3D VHS essentially stems from the critical ring formed in the in-plane direction. With the isotropic in-plane band structure, the dispersion along the critical ring is flat, which leads to the effective dimension reduction of the local band structure. Thus, every momentum on the critical ring can be regarded as a 2D saddle point and presents the logarithmical divergence. That is why we use “critical point” to describe the 3D VHS

in the last version of the manuscript. In the revised manuscript, we further emphasize the presence of the critical ring as the prerequisite of the 3D VHS.

Then, we clarify the energy dispersion around the critical ring and the type of induced VHS. In the mentioned papers [1,2], *Yuan et al.* focused on the VHS in 2D systems and concluded that the energy dispersion around logarithmically divergent saddle point \mathbf{k}_s follows:

$$\nabla_{\mathbf{k}}E = 0 \text{ and } \det D < 0, \quad (1)$$

where $D_{ij} = \frac{1}{2} \partial_i \partial_j E$ is the 2×2 Hessian matrix around \mathbf{k}_s , which can be diagonalized by rotating the axes. The Taylor series of energy dispersion around \mathbf{k}_s in $\mathbf{p} = \mathbf{k} - \mathbf{k}_s$ is $E - E_{\text{VHS}} = -\alpha p_x^2 + \beta p_y^2$. $-\alpha$ and β are eigenvalues of D with $-\alpha\beta = \det D < 0$. This type of VHS is categorized as the ordinary VHS. Meanwhile, they introduce the high-order VHS with the DOS divergence following the power-law. The condition of the high-order VHS formation is defined as:

$$\nabla_{\mathbf{k}}E = 0 \text{ and } \det D = 0. \quad (2)$$

This condition indicates type-I and type-II high-order VHS with both eigenvalues being zero and one eigenvalue being zero, respectively. The energy dispersion around high-order VHS includes at least third-order terms. It is noteworthy that *Yuan et al.* define the type of VHS based on the coefficient of quadratic term obtained from the Taylor series around the momentum of VHS. In contrast, $v_z \geq \sqrt{2}v_{xy}$ is the critical condition in the minimal Weyl model for the formation of 3D VHS. This condition is based on the entire band structure evolution rather than the local energy dispersion around 3D VHS. Under the framework of *Yuan et al.*, we analyze the energy dispersion of the discovered 3D VHS and verify it as an ordinary VHS (due to effective dimension reduction).

Fig. R1 | Comparison between band structure with (without) critical condition satisfied in the isotropic case. a, Energy dispersion along the in-plane direction when $v_z < \sqrt{2}v_{xy}$. **b**, Energy dispersion along the in-plane direction when $v_z \geq \sqrt{2}v_{xy}$. A critical ring appears as denoted by the orange dotted curve. The critical point we

discussed is highlighted by a red dot with tangent q_{tan} and radical direction q_{rad} indicated by white arrows. **c**, Energy dispersion in k_x - k_z plane. The red dot represents the same in **b** with radical q_{rad} and out-of-plane direction q_z indicated by white arrows. The second-order approximations of energy dispersion around the critical point along q_{rad} and q_z directions are plotted in blue and yellow dotted curves, respectively.

When the critical condition is satisfied, the in-plane band structure transforms from a convexity (Fig. R1a) to a Mexican hat (Fig. R1b). With the in-plane isotropy, the band structure is circularly symmetric. The formed critical ring is flat along the tangent direction, as denoted by an orange dotted ring in Fig. R1b, on which each point is equivalent. For simplicity, we focus on the critical point \mathbf{k}_v located at the k_x axis (pointed out by red dot in Fig. R1b) to analyze the divergence behavior of this critical ring. The energy dispersion along in-plane direction is determined by:

$$E_{xy}(\mathbf{k}_{\parallel}) = \pm \sqrt{(\Delta - mk_x^2 - mk_y^2)^2 + v_{xy}^2(k_x^2 + k_y^2)}. \quad (3)$$

We obtain $\mathbf{k}_{v\parallel} = (\frac{\sqrt{v_z^2 - 2v_{xy}^2}}{2m}, 0)$ with $v_z = 2\sqrt{\Delta m}$ by solving $\nabla_{\mathbf{k}_{\parallel}} E_{xy} = 0$.

Importantly, only when $v_z \geq \sqrt{2}v_{xy}$, the critical point is well-defined, which is the origin of the critical condition. Owing to the circular symmetry of the in-plane band, we can check the dispersion around the critical point based on the tangent direction q_{tan} and radical direction q_{rad} of the critical ring, as illustrated in Fig. R1b. They are mutually orthogonal and can be obtained by rotating $k_x - k_y$. The direction perpendicular to the $q_{\text{tan}} - q_{\text{rad}}$ plane is denoted by q_z which is parallel to k_z . Qualitatively, the in-plane energy dispersion around the critical point is flat along q_{tan} and convex along q_{rad} . We examine the energy dispersion utilizing Taylor expansion following *Nat Commun.* 10, 5769 (2019) [1], and the quadratic terms are given by

$$E_v(\mathbf{q}) = -E_{v0} - \alpha q_{\text{rad}}^2 + \beta q_z^2 + \dots, \text{ with } E_{v0} = \frac{v_{xy}\sqrt{v_z^2 - v_{xy}^2}}{2m}. \quad (4)$$

Here, the coefficient factors of quadratic terms are $\alpha = \frac{m}{v} \frac{v_z^2 - 2v_{xy}^2}{\sqrt{v_z^2 - v_{xy}^2}}$, $\beta = \frac{mv_{xy}}{\sqrt{v_z^2 - v_{xy}^2}}$. With

$v_z \geq \sqrt{2}v_{xy}$, the condition $\det D_{E_v} < 0$ is satisfied, indicating that this critical point is an effective 2D saddle point (ordinary VHS) [1], which explains the logarithmically divergent DOS calculated by our band model. The second-order approximation of energy dispersion along q_{rad} and that along q_z are plotted by blue and yellow dotted curves in Fig. R1c, respectively.

We fully agree with the reviewer that the origin of 3D VHS is the critical ring formed in the isotropic in-plane band. This flat critical ring leads to the fact that every point on

the ring becomes an effective 2D saddle point since the energy dispersion along the tangent direction vanishes. Consequently, we assigned this 3D VHS to a critical point in the last version of the manuscript. Following the reviewer’s suggestion, we further analyze the 3D VHS in the revised manuscript utilizing the method from *Nat Commun.* 10, 5769 (2019), and confirm it an ordinary VHS. We also clarify that the critical condition $v_z \geq \sqrt{2}v_{xy}$ is not defined for the local dispersion around the 3D VHS but defined for the energy dispersion of the entire Weyl bands.

Based on your comments, we have revised the manuscript as follows:

- *1. We have added the discussion on the presence of the critical ring and its relation to 3D VHS in the main text Page 11 Paragraph 2.
- *2. We have clarified the type of VHS based on the framework of *Nat Commun.* 10, 5769 (2019) in the Supplementary Section III.

Comment 2:

2. *What is the role of topology in this 3D VHS? Will it protect the 3D VHS?*

Response:

We thank the reviewer for the comment and fully agree that this 3D VHS is not topologically protected. Interestingly, the 3D VHS is connected to the topology of this system. For example, the 3D VHS plays the role of topological Lifshitz transition point when the critical condition is satisfied. For simplicity, we refer to the band structures with/without 3D VHS as “ordinary/VHS case” as illustrated in Fig. R2a and Fig. R2e. We examine the Fermi surface at three characteristic energies E_{F1}, E_{F2}, E_{F3} . At E_{F1} , both cases are in the Weyl semimetal phase with two chiral Weyl pockets as shown in Fig. R2b and f. The topological Lifshitz transition energy E_{F2} corresponds to the zero-momentum energy in ordinary case. In comparison, the 3D VHS serve as the topological Lifshitz transition point in VHS case which divides the Weyl semimetal phase and trivial metal phase. As presented in Fig. R2c and g, two Weyl pockets merge into a single one, letting the net Berry flux on the Fermi surface vanish. An additional Lifshitz transition is expected in the VHS case when Fermi energy is located below E_{F3} (energy of zero-momentum critical point) as the Fermi surface shown in Fig. R2h, while the topology of Fermi surface remains when Fermi energy drops to E_{F3} in ordinary case (Fig. R2d). Therefore, although 3D VHS is not topologically protected, it is tightly associated with the band topology of EuCd_2As_2 .

Fig. R2 | Fermi surface of Weyl bands with (without) critical condition satisfied. **a,** **e,** Energy dispersion of Weyl bands without (with) critical condition satisfied. These two cases are referred to “ordinary case” and “VHS case”. The examined Fermi energies E_{F1}, E_{F2}, E_{F3} are denoted by the green, red and blue planes, respectively. **b-d,** Fermi surfaces of Weyl bands with $v_z < \sqrt{2}v_{xy}$ at different energies. **f-h,** Fermi surfaces of Weyl bands with $v_z \geq \sqrt{2}v_{xy}$ at different energies.

Based on your comments, we have revised the manuscript as follows:

- *3. We have clarified that the 3D VHS is not topologically protected in the main text Page 7 Paragraph 3.
- *4. We have added the discussion that 3D VHS plays the role of the topological Lifshitz transition point in the main text Page 7 Paragraph 3.
- *5. The Fermi surface variation with different energies is provided in Supplementary Section XIII.

Comment3:

3. In fact my answer to question 2 is no. I realize that the in-plane isotropy is crucial for Eq. (1) to produce a critical ring in energy dispersion. If, for example, we apply pressure along one in-plane direction only to break the in-plane isotropy, what will we observe in experiments?

Response:

We agree with reviewer on both points. The VHS is not topologically protected, and the in-plane isotropy is the most crucial factor in producing the divergent VHS.

When in-plane isotropy is broken, the energy dispersion is described by,

$$E(\mathbf{k}) = \pm \sqrt{(\Delta - m_x k_x^2 - m_y k_y^2 - m_z k_z^2)^2 + v_x^2 k_x^2 + v_y^2 k_y^2}. \quad (5)$$

The in-plane anisotropy is determined by m_x/m_y and v_x/v_y . The critical condition will be changed to

$$v_z \geq v_x \sqrt{\frac{2m_z}{m_x}} \text{ and } v_z \geq v_y \sqrt{\frac{2m_z}{m_y}} \quad (6)$$

As shown in the panel iii in Fig. R3a, when both $v_x/v_z \leq \sqrt{m_x/2m_z}$ and $v_y/v_z \leq \sqrt{m_y/2m_z}$ are satisfied, a loop of saddle points is formed (denoted by the orange dashed ring). To distinguish from the critical ring in the isotropic case, we refer to this loop of saddle points as a saddle ring. Notably, this ring exhibits dispersion along its tangential direction, which differs from the flat ring without dispersion in the isotropic case (Fig. R1b). In the anisotropic case, the profile of DOS is mainly determined by the saddle points (denoted by yellow and green dots) located at the extrema of the saddle ring with energies $E_{s1} = v_x \sqrt{4\Delta m_x - v_x^2}/2m_x$ and $E_{s2} = v_y \sqrt{4\Delta m_y - v_y^2}/2m_y$. We individually analyze DOS spectrum with in-plane anisotropy induced by parabolic parameters m_x, m_y and linear parameters v_x, v_y as presented in Fig. R3b. In the isotropic case, the DOS spectrum (black curve in Fig. R3b) exhibits a significant peak with logarithmical divergence which is the origin of our spectroscopic observations. Considering a slightly anisotropic band structure, the DOS spectrum still shows a sharp peak with $v_x/v_y = 0.95$ or $m_x/m_y = 0.95$ due to the negligible energy difference between E_{s1} and E_{s2} . Similar to the isotropic case, peak features are expected to be observed in infrared spectrum with lower intensity. With increased anisotropy, the energy difference between two saddle points increases. Thus, in the DOS spectrum, the sharp peak observed in the isotropic case becomes a broad feature with two kinks corresponding to E_{s1} and E_{s2} . Due to the band broadening caused by thermal effect or impurity scattering, the DOS spectrum of the real system will be much smoother compared to the calculation shown in Fig. R3b. The profile of the optical transition is generally associated with that of DOS. Therefore, the experimental features might be a broad peak, and might even drowned in the spectral background, distinct from the sharp features observed in our experiment.

Owing to the band anisotropy, as illustrated in i panel and iv panel of Fig. R3a, two additional cases of band structure exist with a portion of the critical condition Eq. (6) satisfied. Notably, these two cases are equivalent since k_x and k_y directions can be exchanged. The comparison between DOS spectra of anisotropic and isotropic case is provided in Fig. R3c. The DOS spectrum (grey curve) possesses two kinks stemming from saddle points located at zero and finite momenta denoted by white and orange

points in Fig. R3a, respectively. Compared to the DOS of the isotropic case (red curve in Fig. R3c), the DOS of cases i and iv are much lower and broader, which might be hard to contribute to prominent peak features in spectrum. As for the band structure in panel ii of Fig. R3a, the critical condition is not satisfied at all. Therefore, regardless of whether the band is isotropic, the DOS in this case (blue curve in Fig. R3c) shows a finite kink induced by the 3D saddle point at zero momentum that is fully consistent with the experimental features below the critical field.

Fig. R3 | In-plane energy dispersion with anisotropic in-plane band parameters and corresponding DOS. **a**, Band structure with different strengths of in-plane anisotropy. The energy dispersion along k_x and k_y directions are plotted in red and purple curves. Panels **i** and **iv** show the band structure when energy dispersion along k_x or k_y direction satisfying the critical condition. Two types of critical points, band extremum at zero momentum and saddle points at finite momentum are indicated by white and orange dots, respectively. The band structure in Panel **iii** fully satisfies the critical condition with two types of saddle located at energies E_{s1} and E_{s2} denoted by yellow and green dots. Panel **ii** presents band structure without critical condition satisfied. **b**, DOS spectrum of case **iii** band structure with band anisotropy. The energies of saddle points E_{s1} and E_{s2} are denoted by green and yellow circles, respectively. **c**, Calculated DOS spectrum at different cases. The energy axes in **b** and **c** are normalized by the energy at zero momentum.

In conclusion, we discuss the energy dispersion of the proposed band model with anisotropic band parameters. The critical condition is extended to the formation of a saddle ring. In the isotropic case, due to the flat dispersion along the ring direction, the dimension of the band is effectively reduced, which results in the formation of 3D VHS exhibiting logarithmically divergent DOS (similar to 2D saddle point). This 3D VHS contributes to apparent peak features in magneto-infrared spectra. Regarding the

anisotropic case, the critical ring becomes a saddle ring with finite critical points as discussed above. Given the slight band anisotropy, the energy difference between formed saddle points is negligible, so the DOS spectrum still exhibits an apparent peak with slightly broader width and lower height, leading to spectroscopic phenomena similar to the isotropic case. However, with the anisotropic extent increasing, the DOS peak is further lowered and broadened. Thus, the corresponding feature in the experimental spectrum could be weaker and even drowned in the spectral background.

Based on your comments, we have revised the manuscript as follows:

- *6. We emphasize that (nearly) isotropic in-plane structure is crucial for forming the 3D VHS in the main text Page 11 Paragraph 2.
- *7. The in-plane band structure with anisotropy and corresponding experimental phenomena are introduced in main text Page 11 Paragraph 2 and discussed in detail in Supplementary Section III.

We would like to express our gratitude to Reviewer #1 for encouraging and constructive comments. We have now revised the manuscript with further analysis on the type of 3D VHS and possible experimental phenomena under the case of in-plane anisotropy. It helps to improve the reliability and overall quality of this manuscript. Thank you very much.

Response to Reviewers' Comments

Reviewer 2

General Comment:

This manuscript presents the magneto-infrared spectroscopy results revealing the Van Hove singularity (VHS) in bands of a three-dimensional EuCd_2As_2 under a magnetic field. The authors claim that the VHS appears at magnetic fields greater than 0.6T. The data quality seems to be high, and overall discussions are well written. The concept that the VHS is tuned by magnetic field for the Weyl state bands is very intriguing. If the claim by the author is correct, the results of the discovery of three-dimensional Van Hove singularity would be suitable for publication in Nature Communications. However, I cannot decide whether the current manuscript is suitable for publication or not because of the not-enough presentation. Several concerns described below should be addressed before the judgment.

Response:

We would like to express our gratitude to the reviewer for the valuable and helpful comments. We have thoroughly studied your suggestions and addressed them point by point. For the flat band issue, we clarify that the multiple 4f bands do not contribute to the discussed optical transitions due to the discrepancy in energy positions and linewidth. Meanwhile, the possible signatures of the 4f-bands-related optical transitions are indicated in the high energy range close to the cutoff of our set up. To address your concerns on the determination of optical transitions, we perform the Drude-Lorentz fitting to extract their energy positions and demonstrate that the field-dependent intensity variation agrees with the prediction of two-band model. As for the satellite peaks, we extract the energy positions and analyze their magnetic field evolution. The Kerr rotation is explained by considering different reflectivity of the opposite circularly polarized light. In addition, we provide the raw spectra to support our conclusions (e.g., the existence of $T_\delta, T_\gamma, T_\epsilon$ and energy crossing of T_δ, T_γ). To summarize, we have improved the manuscript in the following aspects:

- 1) Explaining the reason that 4f bands are not the origin of observed optical transition based on the discrepancy in energy positions and linewidth.
- 2) Explaining the reason that the 4f bands are not the origin of satellite peaks for similar reasons.
- 3) Emphasizing the origin of the observed optical transitions based on the magnetic field evolution.
- 4) Clarifying the local flat bands as initial/final states of optical transitions.
- 5) Analyzing the energy dispersion and separation of satellite peaks. Explaining and analyzing satellite peaks based on the Kerr effect.

- 6) Presenting raw spectra to justify the presence of T_α and T_β .
- 7) Performing Drude-Lorentz fitting to strengthen the determination of T_α and T_β .
- 8) Explaining the main peak assignment of T_β based on the Kerr effect and spectral weight distribution among the T_β and its satellite peaks.
- 9) Providing raw spectra to clarify the peak features of $T_\delta, T_\gamma, T_\epsilon$.

Comment 1:

The previous ARPES and DFT calculations [34] demonstrated that the flat bands of EuCd₂As₂ are located deeper than 1eV in binding energy. It is incompatible with the claim by the current authors that it is about 700meV according to the T_β data. The authors should discuss this contradiction.

Response:

We thank the reviewer for the constructive suggestions. The origin of the local flat bands is indeed not clarified in the last version of the manuscript. We agree with the reviewer that multiple flat bands (from the Eu 4f electrons) stay deeper than 1eV below Fermi energy in EuCd₂As₂. We exclude these flat 4f bands as the origin of observed optical transitions (T_α and T_β) because of the discrepancy in the energy and linewidth. On the one hand, 4f bands are nearly continuous in energy due to significant splitting and broadening effects revealed by the previous ARPES result [3] as illustrated in Fig. R4a. The associated optical transitions are expected to generate a very broad spectral feature (nearly 1 eV width) which is generally indistinguishable in our spectrum and differs from the apparent peak features (< 0.1eV width) observed in the experiment. On the other hand, 4f bands are located at extremely deep binding energies, extending up to ~1.75 eV. Most of the related optical responses are expected to emerge far beyond our detectable spectral range. Meanwhile, these flat bands are primarily from the 4f electronic states of Eu, while Weyl bands consist of the electronic states from As and Cd. The limited wavefunctions overlap between these bands could possibly render the corresponding optical transitions forbidden by selection rules.

It is worth clarifying that the “flat band” in our model is different from the general flat band with flat dispersion in the whole Brillouin zone [4]. The momentum position of 3D VHS is located around Γ point and only slightly varies with magnetic fields. Based on the observation of our spectrum, the initial/final states (away from the Weyl bands) should be flat within a very small momentum range near Γ point. Therefore, the related energy bands are not required to be flat in a large momentum range (Fig. R4b). In the revised version, we exclude the origin of 4f flat bands and clarify the “local flat band” to avoid misunderstanding.

Fig. R4 | Previous ARPES result and schematic plot of electronic bands with local flat dispersion. a, ARPES result from Ref. [3]. Flat bands are located at energy deeper than 1 eV with significant energy broadening around 1 eV. Therefore, these electronic states do not contribute to the observed sharp features at the infrared range. **b,** Schematic plot of band structure with “local flat band”. Note that this is a schematic plot, so the scale is not comparable.

Based on your comments, we have revised the manuscript as follows:

- *1. We have clarified in the main text Page 7 Paragraph 1 that the optical transitions are associated with the bands possessing local flat dispersion rather than the flat band through the entire Brillouin zone.
- *2. The reasons for excluding Eu 4f bands as the origin of the observed transition are introduced in the main text Page7 Paragraph 2 and discussed in detail in Supplementary Section VII.

Comment 2:

The authors assume very simple flat bands (one each for the occupied and unoccupied sides in the case without a magnetic field). However, ARPES and DFT calculations demonstrate many of those due to multiple Eu4f levels. Furthermore, these 4f bands split over a wide energy range of about 1eV. The authors should point out this fact in the text and explain why these multiple Eu4f levels should not be considered.

Response:

We fully agree with the reviewer that the discussion on the multiple Eu 4f levels is

highly necessary. In the revised manuscript, we clearly introduce the current understanding and corresponding literature of these multiple Eu 4f levels. We add a more detailed discussion and explain why these multiple Eu 4f levels should not be considered the origin of observed optical features with small linewidths. The main reason is also provided in response to Comment #1. On the aspect of transition energy, 4f bands are located at extremely deep binding energy 1~1.75 eV as demonstrated in Fig. R4a. The corresponding optical transitions are expected to be located at the high energy regime of the spectrum (toward the visible regime). The copper optical pipes and gold-coated focusing mirrors installed in both NHMFL and ECNU setup render the experimental range far below the energy of most 4f-bands-related optical transitions. In fact, we found a pronounced spectral feature around 1300 meV as highlighted by red arrows in Fig. R5 which might serve as the possible signature of the optical transition originating from 4f bands. On the aspect of linewidth, 4f bands exhibit significant energy broadening of 1 eV. It contradicts the small width of observed optical transitions. Therefore, 4f bands are excluded as the possible origin. For the same reason, the satellite peaks alongside the T_β are confirmed not from the multiple 4f bands.

Fig. R5 | Magneto-infrared spectroscopy in NIR regime. **a, b,** magneto-reflectivity R_B/R_0 measured at 8 K and 2 K. Constant offset is applied for clarity. A noteworthy peak around 1300 meV (pointed by red arrow) might serve as the signature of the optical transition associated with Eu 4f levels due to the large energy.

Instead, we assign T_α and T_β to the optical transitions between Weyl bands and other higher/lower-energy electronic bands with local flat dispersion around Γ point (depicted in Fig. R4b) based on their field-dependent evolution in the magneto-mid-infrared spectra. Importantly, our model further predicts the optical transitions between

these local flat bands (i.e., T_γ and T_ϵ). These predictions are verified in the magneto-near-infrared spectra, which further consolidates the model and excludes the contribution of 4f bands.

Based on your comments, we have revised the manuscript as follows:

- *3. We have introduced both the current understanding and corresponding literature of these multiple Eu 4f levels in the main text Page 7 Paragraph 2.
- *4. We have explained why multiple Eu 4f levels are excluded as the origin of optical transitions in the main text Page 7 Paragraph 2 and Supplementary Section VII.
- *5. The spectra with possible signatures of 4f-band-related optical transitions are presented and discussed in Supplementary Section VII.

Comment 3:

The authors attribute the satellite peaks to the Kerr effects but do not explain why they believe so. The data quality of Fig.2d looks high enough to extract the detailed peak positions and their evolution with an external magnetic field. I request that the authors show not only color maps but also spectra, and explain the energy dispersion, their separation, and how to understand those based on the Kerr effects in the main text.

Response:

This is a good point. We extracted energies of observed satellite peaks to analyze their field evolution and separation. As shown in Fig. R6a, the satellite peak series can be divided into two groups according to relatively strong and weak intensity (hereafter referred to as “strong peak series” and “weak peak series”). The stacking plot of the spectra is shown in Fig. R6b where the satellite peaks can be visualized. The energies of all strong and weak peaks are extracted in Fig. R6c, as denoted by hollow squares and solid diamonds, respectively. The separation between adjacent peaks in each series becomes narrower at higher energy. Above ~ 600 meV, the separation between the strong peak series is too narrow to distinguish the weak peak series. Each extracted peak follows the linear scaling (solid curves with light color in Fig. R6c) with 8 K magnetization as exhibited in Fig. R6d.

Fig. R6 | The satellite peaks in mid-infrared spectroscopy. **a**, False-color plot of magneto-infrared spectra in mid-infrared regime. The satellite peaks are distinguished and categorized into two series based on intensity. **b**, Stacking plot of magneto-reflectivity spectra up to 12 T with a constant offset for clarity. **c**, The extracted energies of satellite peaks denoted by hollow squares (for the strong series) and solid diamonds (for the weak series). The energy variation of satellite peaks with fields is consistent with the linear scaling of the magnetization. The linear scaling curves are presented as solid curves with lighter colors. **d**, The magnetization at different temperatures.

In the previous Kerr/Faraday rotation research [5–7] of the magnetic materials and spin-polarized systems, the smoking-gun phenomenon is a series of oscillating satellite structures with the energy period decreasing at higher energy and terminating at the highest-energy feature related to the corresponding inter-band transition. As shown in Fig. R7, the local minima are indicated by grey dashed lines. The spacing between adjacent minima becomes narrower at higher energy, which agrees with previous reports. Under external magnetic fields, electronic bands of EuCd_2As_2 become spin-polarized due to exchange interaction. Therefore, this system exhibits different reflectivity for left-handed (LCP) and right-handed circularly polarized (RCP) lights. The difference between LCP and RCP reflectivity depends periodically on the wavelengths and this effect depends linearly on the magnetization. Therefore, the combined reflectivity exhibits an oscillating satellite structure. Importantly, the energy dispersion of satellite peaks as well as their energy separation are expected to follow the linear relation on the magnetization instead of magnetic fields which is fully verified by the experiments. The oscillating phenomenon could be amplified if two linear polarizers parallel (perpendicular) to each other are mounted in the incident and exit light paths [5]. The consistency between experiments and expectations is reflected in the following facts: (i) the presence of satellite structures, (ii) linear dependence on the magnetization, (iii) decreasing period at higher energy, (iv) satellite peaks merging and terminating at the highest-energy feature from the inter-band transition. Thus, the Kerr rotation is identified as the most possible origin of the observed satellite peaks. It also explains the absence of satellite peaks alongside the transition T_α since the initial state of T_α splits negligibly.

Fig. R7 | The magneto-reflectivity spectrum measured in ECNU. A series of satellite peaks can be visualized. The separation between adjacent minima (indicated by grey dashed lines) decreases at higher energy. The satellite features merge and terminate at around 630 meV.

Based on your comments, we have revised the manuscript as follows:

- *6. We have provided spectra in Supplementary Section IV.
- *7. We have added the discussion on the origin of satellite peaks in main text Page 8 Paragraph 2.
- *8. We have added a detailed discussion on the Kerr effect and extracted the field evolution (dispersion and separation) of satellite peaks in Supplementary Section V.

Comment 4:

I can understand that the peak positions of T_{β} can be defined precisely from the data in Fig.2d but I wonder if it is possible for T_{α} . More seriously, I suspect that T_{α} and T_{β} could not be determined from Fig2c with poor quality. In particular, the error bar is not indicated (or negligible) in Fig.2e. These suspects come from the not-enough presentation of the authors plotting only color images, not spectra. I request that the authors show spectra, not only images (Fig.2c,2d), and demonstrate how they determined T_{α} and T_{β} .

Response:

Thank you for the suggestions regarding the determination of optical transitions. We provide the spectra and conduct the Drude-Lorentz analysis to elucidate the identification of optical transitions. In the NHMFL spectra, we point out the apparent

peak features corresponding to T_α and T_β by blue and red arrows in Fig. R8a, respectively. The signal-to-noise level of NHMFL spectra is comparatively lower owing to the much lower detectivity of bolometer in the high-energy regime (> 0.5 eV). That leads to the lower intensity and broader linewidth of spectral features. Therefore, these optical features are better resolved in the false-color plot (Fig. R8c) where broad features are usually better visualized. In the original manuscript, we extract the positions of apparent features in the false-color plot. This is also the reason why we repeat the experiments in the ECNU setup with better detectivity in the mid-infrared regime (Fig. R8b, d).

Fig. R8 | Magneto-reflectivity spectra up to 12 T in mid-infrared regime measured in NHMFL and ECNU. a, b, Stacking plots of magneto-reflectivity spectra measured at NHMFL and ECNU. Constant offsets are introduced for clarity. The peak features of T_α and T_β are indicated by blue and red arrows, respectively. c, d, False-color plots of magneto-reflectivity spectra measured at NHMFL and ECNU. The spectra features of T_α and T_β are pointed out by the blue and red arrows, respectively.

As for the ECNU data illustrated in Fig. R8b, T_β and its satellite peaks can be well determined because of the much lower noise level compared to the NHMFL data. The T_α overlaps with satellite peaks at high fields and those satellite peaks dominate the spectrum due to much narrower width and larger amount of features. Once T_α overlaps with satellite peaks, it will be more challenging to be directly visualized but still can be resolved in the stacking plot (Fig. R8b). Once again, such broad features with overlapping conditions are naturally better resolved in the false-color plot (pointed out by blue arrows in Fig. R8d).

To better address the reviewer's concerns on peak identification and energy extraction, we performed Drude-Lorentz fitting to reproduce the spectrum and extract the corresponding peak parameters. The dielectric function Drude-Lorentz model reads [8,9]

$$\epsilon(\omega) = \epsilon_{\infty} - \sum_i \frac{\omega_{p,D;i}^2}{\omega^2 + \frac{i\omega}{\tau_{D,i}}} + \sum_j \frac{\Omega_j^2}{\omega_j^2 - \omega^2 - i\omega\gamma_j}, \quad (6)$$

where ω is the frequency; ϵ_{∞} is the real part at high frequency; the first sum denotes free carrier response with plasma frequency $\omega_{p,D;i}$ and scattering rate $1/\tau_{D,i}$ of the i th Drude component; the second sum denotes the bound excitations with strength Ω_j , energy position ω_j and linewidth γ_j of the j th Lorentz component. We transform the dielectric function to reflectivity in order to fit with the experimental spectrum. The fitting function includes all related optical excitations which helps to resolve the overlapping issues. It is worth noting that the analysis here is primary since it may take special caution while dealing with relative magneto-reflectivity spectra with the Drude-Lorentz function. Here, we plot the typical fitting results at 7 T that fits the experimental results fairly well as shown in Fig. R9a. The fitted energies of optical transitions are located very closely to the apparent peak positions on the reflectivity spectra. Thus, it is reasonable to extract the transition energy by the apparent peak position. The energy uncertainty associated with these optical transitions has been added to the graph. Moreover, we try to theoretically reproduce the entire magneto-infrared spectra based on (i) scaling parameters (linear relation between peak energy and magnetization) extracted from experiments, (ii) the DOS predicted by the two-band minimal model, (iii) the Drude-Lorentz function. As shown in Fig. R9c, we successfully reproduce the entire spectra, which validates the overall peak identification and energy extractions. It is important to note that the theoretical reproduction cannot fit with experiments without the inclusion of Lorentz function from T_{α} .

Fig. R9 | Magneto-reflectivity spectra measured in ECNU and reproduced by the Drude-Lorentz function. a, The experimental spectrum at 7 T and the Drude-Lorentz fitting result. An offset is applied on the fitting curve for clarity. **b**, False-color plot of spectra measured in ECNU. **c**, False-color plot of theoretical spectra reproduced by the

Drude-Lorentz function based on the scaling parameters extracted in experiments and DOS calculated from the two-band model.

Based on your comments, we have revised the manuscript as follows:

- *9. The stacking plots of the spectra have been provided in Supplementary Section IV.
- *10. The Drude-Lorentz fitting results and the peak identification have been added in Supplementary Section IV.
- *11. Figs. 2e-f have been revised with error bars.
- *12. We have added the discussion on the relation between false-color plots and stacking plots of spectra in Supplementary Section IV.
- *13. The theoretical reproduction of the magneto-infrared spectra and justification of peak determination are provided and discussed in Supplementary Section IV.

Comment 5:

Related to the above, it was unclear why the authors assigned the highest energy of T_{β} as the main peak and the others as satellites. As mentioned above, Eu4f levels are possibly the reason for these many peaks.

Response:

We thank the reviewer for the suggestions. As discussed in response to Comment #2, the multiple 4f bands have been excluded as the possible origin of the multiple peak features observed experimentally. The main reason is the discrepancy in the energy scale and band broadening.

Comparing our experimental results with previous research, we consider the Kerr effect for understanding multiple peak features and set T_{β} as the main peak. On the one hand, the T_{β} is identified as the main peak due to its much sharper intensity compared to the satellite component. This is better visualized in both false-color plot and spectra measured in the ECNU setup as shown in Fig. R8b&d. On the other hand, among satellite peaks, T_{β} is the optical transition with the highest energy where oscillating components merge and terminate. This is the typical spectral feature of the Kerr/Faraday effect on magnetic materials and spin-polarized systems [5–7], owing to the different reflectivity of opposite circularly polarized lights. The resultant polarization variation of the reflected light induces oscillating satellite features in the magneto-infrared spectrum. With energy increasing, the oscillation period is expected to drop and finally merge with the main feature related to inter-band transition, which corresponds to the observed T_{β} with the highest energy in our experiment. Consequently, T_{β} with the sharpest feature and the highest energy is assigned as the main peak based on the consistency with the Kerr effect.

Based on your comments, we have revised the manuscript as follows:

- *14. The discussion on the assignment of T_β has been added to the main text Page 8 Paragraph 2 and Page 6 Paragraph 1. The associated analysis has been presented in Supplementary Section V.

Comment 6:

The ECNU data are of much higher quality than the NHMFL data. The only advantage of the ECNU data is to double-check their conclusion. In that case, I request that the authors move the NHMFL data to supplemental information, and comment only briefly in the main text that the conclusion is reproduced by another data set.

Response:

We thank the reviewer for the suggestion and fully agree that the ECNU data exhibits much higher quality. In light of the reviewer's suggestions, we removed the NFMFL data (original main text Fig. 3a) to Supplementary Section X.

It is highly unusual in magneto-infrared spectroscopy that spectral intensity abruptly increases through such a wide energy range (hundreds of meV). It might be induced by the instrumental problem around high field conditions. So, it is of importance to double-check the results with independent setups. Thus, we move the low-field data to Supplementary Information and keep the plot in Fig. 2c. Please let us know whether you find the arrangement appropriate. Thank you.

Based on your comments, we have revised the manuscript as follows:

- *15. We have removed the low-field spectra measured at NHMFL to Supplementary Section X and introduced them briefly in the main text Page 6 Paragraph 1.

Comment 7:

The authors claim that T_δ and T_γ cross but it is unclear to me. A better presentation using real spectra (not only color plots Fig4c) is necessary to convince the readers. I request that the authors plot spectra, not only color images, and justify this claim.

Response:

We fully agree with the reviewer that real spectra help to resolve the cross. We present the raw spectra in the near-infrared regime as shown in Fig. R10. The peak features of T_δ and T_γ are traced by pink and green triangles, respectively. The crossing phenomenon is more evident in spectra at 2 K (Fig. R10b) than in 8 K spectra (Fig.

R10a) due to sharper magnetization in 2 K (Fig. R6d) and reduced thermal broadening effect.

Fig. R10 | Magneto-reflectivity spectra measured in near-infrared regime. a, b, Stacking plots of magneto-reflectivity at 8 K and 2 K. Peak features corresponding to T_δ , T_γ and T_ϵ are denoted by pink, green and blue triangles, respectively. Notably, different offsets are introduced in the two panels of **a** to make all peak features distinguishable.

Based on your comments, we have revised the manuscript as follows:

*16. The 2 K and 8 K spectra have been provided in Supplementary Section XI.

Comment 8:

Related to the above, T_γ , T_δ , and T_ϵ are all unclear to me. To justify their claim, the author should display spectra not only color images for Fig4c, and justify their claims that these are indeed obtained from the data.

In conclusion, I will reconsider whether the data are sufficient to meet the authors' claims after the major revision.

Response:

To address your concerns on the existence of T_δ , T_γ and T_ϵ , we provide the raw spectra with all transitions traced as shown in Fig. R10. According to the apparent peak height, T_γ and T_ϵ are weaker than T_δ . It agrees with the theory prediction because T_γ and T_ϵ originate from the electronic bands with local flat dispersion rather than 3D VHS with divergent DOS. Above 1.6 T, the peak of T_ϵ starts to overlap with the prominent excitation around 1300 meV which may originate from the 4f-bands-related optical transition as discussed in response to Comment #1.

Based on your comments, we have revised the manuscript as follows:

- *17. The 2 K and 8 K spectra have been provided in Supplementary Section XI with peak features of T_δ , T_γ and T_ϵ highlighted by corresponding symbols.

We would like to express our sincere appreciation to Reviewer #2 for raising important questions on the role of multiple 4f bands, optical transition determination, Kerr effect and peak assignment. We fully recognize that the corresponding discussion and data presentation are important for understanding the magneto-infrared spectroscopic result of EuCd_2As_2 . With additional Drude-Lorentz analysis, spectra reproduction and sufficient data presentation, our conclusion is further strengthened. We hope that the revised manuscript has been improved to address all your concerns. Thank you very much for your constructive and valuable input.

Response to Reviewers' Comments

Reviewer 3

General Comment:

Wu et al. present a magneto-optical study on EuCd₂As₂ up to 12 T, primarily at low-temperature AFM phase. The authors proposed a model for their observations based on the appearance of a 3D VHS in this material. The request for this model is that the system is a Weyl semimetal and the nodes can be tuned by the external magnetic field. The Weyl semimetal nature of this compound is widely accepted, especially based on ARPES measurements that probe the band structure and report Dirac nodes in the vicinity of the Fermi energy. However, one should point out that these measurements are more surface-sensitive.

The idea of 3D VHS in a real system is exciting, especially in a system that can be easily tunable with an external magnetic field. However, to my opinion, here authors try to explain their results with biased knowledge and at certain points fail to provide a full picture of the observations. Therefore, I cannot recommend this work for publication in Nature Communications without a more thorough examination of the presented data. Here are some of my concerns that the authors should consider:

Response:

We would like to thank the reviewer for the critical and constructive comments. In particular, the comments raised by Reviewer #3 regarding the semiconductor gap observed in recent work (including the self-doping effect) really help us reach a more comprehensive understanding of the electronic structure in EuCd₂As₂ system. In the revised version, we proposed a more realistic band model which is generally consistent with both the energy dispersion directly probed by ARPES and the various phenomena in resistivity as well as optical spectrum. The insulating and metallic samples can be explained by further considering the shifted Fermi level determined by impurity condition and the self-doping effect. To address the concerns on spectral evolution near the critical field, we present the raw data and demonstrate that the spectral behavior agrees with the theoretical prediction. Regarding the origin of T_α , we exclude the cyclotron resonance by analyzing its field evolution. To summarize, we have improved the manuscript in the following aspects:

- 1) Presenting raw spectra to verify the field evolution of T_α and T_β ;
- 2) Analyzing the peak features of T_β that can be traced below the critical field and comparing their intensity with theoretical DOS;
- 3) Explaining the reason for excluding Weyl nodes as the origin of the observed

optical transitions. We further clarify that the presence of Weyl nodes is not prerequisite for the formation of VHS.

- 4) Presenting resistivity data of different samples and excluding the cyclotron resonance as the origin of T_α .
- 5) Proposing a more realistic band model that is consistent with the ARPES/transport/optics results of both metallic and insulating samples. This band model potentially resolves the controversy in EuCd_2As_2 and might provide a general understanding of both our result and previous research utilizing various techniques.

Comment 1:

Authors claim that there is a critical magnetic field of ~ 0.6 T initiates certain optical transitions that are in principle at the heart of the discussions. However, a quick inspection of Fig2 a and b reveals that the $T\beta$ transitions in fact extend all the way to the zero field (with MCT detector, obviously bolometer was not sensitive enough), whereas the $T\alpha$ transitions cannot be traced because the lowest energy that the authors presented is 300 meV that is quite large in principle. Therefore, it is not clear if the features evolve gradually or appear at so called Bc.

Response:

Thank you for the comment. To clarify the above issue, we exhibit the stacking plot of raw spectra measured in both setups and demonstrate that the experimental spectra agree with the theoretical density of states (DOS). Fig. R11a and b exhibit raw spectra measured in ECNU (using our home-built magneto-infrared setup) and NHMFL, respectively. The peak features of T_α are traced to low fields with pink triangles. In both ECNU and NHMFL data, the peak feature of T_α is very weak below the critical field 0.6 T. An abrupt height increase of T_α peak can be found around 0.6 T, which in our model, is related to the formation of the 3D Van Hove singularity (VHS).

The NHMFL spectra present higher data quality below 300 meV (due to the bolometer detector and lower cutoff frequency in the setup) and prove the absence of optical features with field-dependent energy in that frequency range. Since the spectra above 300 meV are enough to show the complete field evolution of the optical transitions according to the raw data, we set it as the lowest energy of the plot. In much higher magnetic fields than 0.6 T, T_α can not be easily traced in the stacking plot due to the energy overlap with satellite peaks. The satellite peaks dominate the spectrum behavior owing to much narrower linewidth. It also explains why the T_α can be better resolved in the false-color plot where broad features are generally better visualized. For the same reason, T_α becomes better resolved in the stacking plot with the worse signal-to-noise ratio (mid-infrared regime in NHMFL).

Fig. R11 | Magneto-infrared spectra measured at ECNU and NHMFL. a, Magneto-infrared spectra measured in ECNU. The peak features of T_α are traced to 0.4 T. **b,** Magneto-infrared spectra measured in NHMFL with higher spectral quality below 300 meV compared to ECNU data. Constant offsets are applied for clarity.

We agree with the reviewer that T_β can be traced below the critical field. In fact, this phenomenon is well predicted by our model. Based on the proposed model, the overall DOS is contributed by both critical points in zero momentum (non-divergent) and finite momentum (VHS). The energies of these two types of critical points merge at B_c . The former critical point presents a prominent kink in the DOS, leading to observable T_β below B_c . After reaching B_c , the VHS forms. Consequently, the induced DOS are expected to increase much more sharply than the case before B_c due to the divergent nature of VHS. The sharp increase of the peak eventually terminates around the magnetic saturation field B_s . Therefore, the model predicts a “3-stage” intensity variation: being low but observable below B_c ; increasing sharply after reaching B_c ; becoming saturated after the saturation field of magnetization B_s . The observed intensity variation agrees well with the model prediction. In the revised manuscript, we add more discussion on the “3-stage” intensity variation and the system response below B_c .

We further try to analyze the peak intensity more quantitatively based on the minimal model. We start from the satellite peaks due to their well-defined peak dip features. The spectral weight of T_β is largely assigned to the satellite peak. Meanwhile, the field dependence is almost identical between T_β and its satellite peaks. We note that the following analysis is preliminary because it may take additional caution while dealing with the peak height in the magneto-reflectivity spectra.

We choose several well-defined satellite peaks (indicated by arrows in the left panel of Fig. R12a) and extract the apparent peak positions and heights to track the intensity variation of field-induced optical transitions. As shown in Fig. R12b, peak heights of all satellite peaks follow “3-stage” behavior and prominently rise around the critical field. Before B_c , the peak heights are finite and increase gradually. To compare with the proposed model, we extract the peak height of the predicted DOS from the theory (Fig. R12c). Here, we assume that the transition intensity variation generally reproduces the behavior of DOS, and transition matrix elements roughly remain constant since the magnetic field seems to only shift the related band without significant field-dependence of the band itself. The upper panel of Fig. R12c shows the DOS variations with and without energy broadening (red and blue curves) caused by thermal effect or disorder scattering. Broadening energy is set as 0.5 meV in the calculation. With or without the effect of broadening, the calculated DOS reproduces the “3-stage” behavior and agrees well with the experimental observation.

Fig. R12 | Comparison between apparent peak height of transition features and that of DOS peak. a, False-color plot of magneto-infrared spectra measured at ECNU. To illustrate the spectral variation around the critical field, we select four series of peaks (denoted by arrows in the left panel) with their energies presented in the right panel. **b,** Extracted apparent peak height H_{R_B/R_0} of selected four series of peaks from magneto-reflectively spectra R_B/R_0 . **c,** Apparent peak height of calculated DOS H_{DOS} normalized by the maximum value. The calculation with (without) energy broadening is denoted by the red (blue) curve. Zoom-in presentation within 2 T is shown in the bottom panel. Sharp increase around B_c can be better visualized.

Based on your comments, we have revised the manuscript as follows:

- *1. Raw spectra measured in ECNU and NHMFL have been added in Supplementary Sections IV and X to clarify the field evolution of optical transition and briefly introduced in the main text Page 6 Paragraph 1.
- *2. We have revised the description of the optical transition evolution around the critical field throughout the main text, such as Page 4 Paragraph 3, Page 5 Paragraph 3, Page 6 Paragraphs 1&3 and Page 13 Paragraphs 2&3.

- *3. We have emphasized the “3-stage” intensity variation observed in our magneto-infrared spectrum and theoretical predictions in the main text Page 6 Paragraph 2, Page 7 Paragraph 3 and Page 8 Paragraph 1.
- *4. The apparent heights of the typical peaks have been extracted and presented in Supplementary Section X.
- *5. The calculated DOS spectra have been added in Supplementary Section X. We further discuss the consistency between experiment and theory in the main text Page 9 Paragraph 1.

Comment 2:

Authors never show a raw spectra that would allow one to assess the spectra by itself in terms of signal-to-noise ratio. I think it is important for the previous point, as well.

Response:

We fully agree with the reviewer that raw spectra are helpful in identifying the spectra feature and understanding the field evolution of the optical transitions. In Fig. R13, we provide the stacking plot of raw data measured in both the NFMFL and ECNU setups. Spectra are vertically shifted with a constant offset for clarity. The optical transitions in the raw data exhibit a “3-stage” variation: being low but observable below B_C ; increasing sharply after reaching B_C ; becoming saturated after the magnetic saturation field of magnetization B_S . T_β is much better resolved in ECNU data because bolometer detector efficiency is really low at energy higher than 500 meV.

Fig. R13 | Stack plot of magneto-infrared spectra measured in NFMFL and ECNU. The spectra are vertically shifted with a constant offset.

Based on your comments, we have revised the manuscript as follows:

- *6. The raw spectra have been added to Supplementary Section IV with the corresponding discussion on the “3-stage” intensity variation.

Comment 3:

Authors discuss basically the optical transitions between saddle points arise due to the band crossings and gradual move of these points with magnetic field. Looking to the band structure of EuCd₂As₂ one would expect these crossings at the very low energy region. However, authors present transitions located at quite a large energy regime. How these are directly related with the low energy features of the Weyl nodes?

Response:

This is a very good point. In fact, the optical transitions observed in our spectrum originate from the excitations between the Weyl bands and other bands (at much higher or lower energy) with local flat dispersion around Γ point rather than the excitation between saddle points. Consequently, the energies of these optical transitions can reach hundreds of meV, which significantly exceed the energy of transitions within the Weyl nodes (typically within 100 meV) as reported in previous research [10–13].

It is indeed crucial to emphasize that the optical transitions are associated with the higher-energy part of Weyl bands (around saddle point) rather than the lower-energy part (around Weyl nodes). The observed spectral features, especially the apparent intensity enhancement at the critical field, can be understood with the field evolution of energy bands and the formation of 3D VHS as described by the minimal model, while the low-energy properties of Weyl nodes are not included. Although the Weyl minimal model is used to capture the main physics, it is important to note that the discussed VHS physics and related experimental findings remain valid even if the low-energy nodes are gapped, or the Fermi level is away from the low-energy crossing. Instead, the key point is the formation of a critical ring. This is also pointed out by Reviewer #1. Therefore, the Weyl semimetal phase is not a prerequisite for accessing the 3D VHS. In the revised manuscript, we emphasize this point twice for better readability. This fact is also one of the reasons that we describe the EuCd₂As₂ as a “topological magnet” rather than “magnetic Weyl semimetal” in our manuscript.

Although the Weyl semimetal phase is not required, the two-band minimal model is one of the simplest models to describe the discussed VHS physics (around the saddle ring). The model is also consistent with the possible general understanding of the compound as further discussed in response to comment #5.

Based on your comments, we have revised the manuscript as follows:

- *7. We have clarified the origin of observed optical transitions and emphasized that the low-energy Weyl nodes are not included in the physical explanation in the main text Page 7 Paragraph 1 and Page 11 Paragraph 2.
- *8. We have emphasized twice on the “low-energy Weyl nodes are not the prerequisite for forming the 3D VHS” in main text Page 11 Paragraph 2 and Supplementary Section III.

Comment 4:

Authors does not provide any resistivity curves for their sample, but considering that is available in the literature one would assume that they have a seemingly metallic compound. In this case how the cyclotron motion of the free carriers are coming to the discussion? Could it be that T_α is in fact this cyclotron frequency rather than some optical transition?

Response:

We thank the reviewer for the insightful comment on the transport properties of our samples and the origin of T_α .

The transport property is generally determined by the electrons near the Fermi level. In our magneto-infrared spectra of EuCd_2As_2 , those optical responses with field-dependent features are located at quite high energy (several hundreds of meV). Meanwhile, the model predictions are also of high energy and insensitive to the position of Fermi energy. Therefore, the DC resistivity is not directly linked to the main finding of this manuscript. On the other hand, we indeed studied the transport properties of the as-grown crystal but found the results highly complex when compared with previous transport studies [3,14–16]. For example, our sample exhibits insulating behavior as evidenced by the resistivity vs. temperature curve in Fig. R14a, distinct from the typical metallic behavior reported in previous research [14] as shown in Fig. R14b. We are also surprised to find that the resistivity of our sample increases several orders of magnitude after its surface is polished, which indicates that there may exist unconventional surface contributions to transport channels. The transport properties of EuCd_2As_2 system appear to be determined by multiple origins and depend on the details of the surface condition.

Meanwhile, self-doping is another critical factor for the transport property of EuCd_2As_2 . Most of the reported metallic samples manifest the prominent hole-doping revealed by ARPES result [3] and Hall measurement [17,18]. The insulating sample with a much slighter doping level can be obtained by utilizing the two-step flux growth method and

improving the purity of starting materials [19]. Here, we would like to emphasize that the transport property is mainly determined by the Fermi surface carriers, so transport results are very sensitive to the position of Fermi energy, impurity condition and the self-doping effect. In comparison, the high-energy optical transitions we observed mainly reflect the field evolution of high-energy bands in EuCd_2As_2 which is not much influenced by the energy position of Fermi level. Although the transport exhibits complex variation with self-doping effect, synthesis method, or possible surface contribution, the discussed VHS physics described by the minimal model should not be strongly influenced. Our optical spectra and model remain generally invariant if a large gap is generated between the VHS and local flat bands. We have added resistivity data in the revised version and emphasized that the discussed VHS physics does not directly relate to the metallicity or insulativity of the sample.

In response to Comment #5, we further extend our toy model to a more realistic band model (Fig. R15a) that potentially provides a universal understanding of experimental facts from both metallic and insulating samples utilizing various techniques.

Fig. R14 | Temperature-dependent resistivity of our sample and reported in previous research [14]. a, Resistivity vs. temperature curves of our sample measured before and after surface polish. Compared to the pristine sample, the resistivity increases several orders of magnitude in the polished one. It indicates possible unconventional surface contribution in our sample. Despite the complexity of the resistivity, it is not directly related to our spectral finding and discussed VHS physics. **b,** Resistivity vs. temperature curves from Ref. [14].

As for the possible cyclotron resonance origin of T_α , we exclude it by analyzing its energy and intensity variation with magnetic fields. On the aspect of energy, cyclotron resonance energy is proportional to the magnetic fields for Schrödinger fermions (might be different for other types of quasi-particles, but generally positively correlated with

magnetic field). However, the observed energy of T_α increases with magnetic fields at low field range but becomes fully saturated after the saturation field of magnetization B_s , clearly following the magnetization. As for the intensity evolution, the spectral weight of the cyclotron resonance is expected to continuously enhance with the magnetic field because the degeneracy of Landau levels is proportional to the external field, regardless of the band dispersion. In contrast, T_α transition undergoes a significant amplification at the critical field but becomes stationary above the magnetic saturation field B_s . A critical clue is that the variation of T_α almost linearly depends on magnetization but not magnetic field. It helps to exclude the origin of cyclotron resonance and points to the electronic states shifted by the exchange interaction.

Based on your comments, we have revised the manuscript as follows:

- *9. We have added the temperature-dependent resistivity data and discussed the origin of the various transport properties of EuCd₂As₂ in Supplementary Section IX.
- *10. We have clarified that the observed high-energy optical transitions are not sensitive to the Fermi level compared to the transport property in the main text Page 12 Paragraphs 1&2. We have emphasized that the discussed VHS physics does not depend directly on the metallicity or insulativity of the sample.
- *11. We have compared the field evolution of T_α with typical cyclotron resonance behavior and further exclude it as the origin of T_α in the main text Page7 Paragraph 1.

Comment 5:

Finally, I would like to make a general remark on this compound. Although the Weyl semimetal picture seems to be widely accepted widely accepted situation for EuCd₂As₂, it is recently challenged with the optical studies itself (ArXiv:2301.08014). In fact, these are magneto-optical studies up to 16 T and directly related to this study, which is never mentioned by the authors in the manuscript in any context.

I should separately emphasize that the work at the last point is not peer reviewed and therefore I don't intend to make the authors accountable for these results, but it is in fact very important work because it directly provides an observation of the bulk properties of the material and gives evidence that the impurity effects are extremely important for this compound. Whereas the self doping effects could render the resistivity metallic, overall bulk material is a semiconductor with the band gap of on the order of ~700 meV. With magnetic field it is reduced but never closes suggesting that the band crossings creating Weyl nodes do not exist.

Response:

We thank the reviewer for bringing up the infrared study of the same system reported by Santos-Cottin *et al.*, which is preprinted at *ArXiv:2301.08014* and now published at *Phys. Rev. Lett. 131, 186704 (2023)* (refers to “PRL work” hereafter) [19]. We became aware of it while our manuscript was under review. The reviewer’s comments on the self-doping effect and band structure (especially band crossing) are very insightful. In the following, we demonstrate that our findings actually agree with the PRL work in many aspects. In the revised manuscript, we have added discussions on the related literature (semiconductor nature, self-doping effect, etc.). By further extending the minimal model to a more realistic version, we raise a possible understanding that might resolve the controversy of EuCd_2As_2 .

The main findings of PRL work are summarized as follows. Santos-Cottin *et al.* synthesize an insulating sample by purifying the seed crystals obtained from the first flux growth, while samples in most previous research are metallic. Combining with multiple experimental techniques including pump-probe angle-resolved photoemission spectroscopy (ARPES) and magneto-infrared spectroscopy, they challenge the Weyl semimetal phase in EuCd_2As_2 based on the observations: (i) Transport property of EuCd_2As_2 is determined by the hole-doping level; (ii) The ARPES result along in-plane direction exhibits a band gap ~ 770 meV; (iii) The magneto-infrared spectroscopy indicates that a dip feature undergoes energy shift of ~ 100 meV. Based on the above experimental findings, it is self-consistent for them to adopt a semiconductor picture of EuCd_2As_2 . Although we adopt the Weyl minimal model to describe the VHS physics, as concluded in our response to Comment #4, it does not require the EuCd_2As_2 to be a Weyl semimetal for accessing 3D VHS. This is mainly because the VHS physics and the observed transition are not related to the low-energy band crossing. Instead, the saddle ring forming along the in-plane direction plays a crucial role in forming 3D VHS. Even if the Fermi energy is away from band crossing or even a large gap is generated, the 3D VHS physics remains valid. Therefore, our finding does not contradict the PRL work.

Meanwhile, EuCd_2As_2 is widely recognized as Weyl semimetal, and this fact should also be seriously considered. It is reasonable to shift the Fermi energy or doping level by different growth methods, however, the Weyl bands should not simply change to trivial dispersion while doing so. Therefore, we carefully study the comprehensive ARPES report which directly demonstrates the presence of Weyl nodes [3] (*Sci. Adv. 5, eaaw4718 (2019)*, refers to “SA work” hereafter). By summarizing our findings and both results of PRL work and SA work, we extend the toy model of Weyl semimetal and propose a more realistic band structure of EuCd_2As_2 as illustrated in Fig. R15a.

It is worth noting that the symmetry-protected band crossings (possible Weyl nodes) are only allowed to exist along the $\Gamma - A$ direction (k_z direction) according to the theoretical analysis in previous research [20–22]. Consequently, the signature of the Weyl bands can not be captured in the ARPES focused on the in-plane energy dispersion around Γ point. The *PRL* work presents the ARPES results along the in-plane direction (k_{\parallel}) as shown in Fig. R15b, the reported absence of Weyl nodes is reasonable and agrees with the in-plane results of *SA* work (Fig. R15c). These results also support the hole-doping level as the key factor for the transport properties of EuCd_2As_2 . However, the ARPES result of energy dispersion along the k_z direction provided in *SA* work verifies the existence of Weyl bands as shown in Fig. R15d. The results from *PRL* and *SA* work along different directions actually agree well with the general understanding of EuCd_2As_2 as well as our toy model.

In addition, an energy gap is found in *PRL* work based on the pump-probe ARPES and infrared spectra. In fact, this finding also agrees with both our results and *SA* work. In our magneto-infrared spectra, the observed large energy of T_{β} indicates the absence of electronic structure within ~ 700 meV above the VHS. For *SA* work, static APRES is only sensitive to the occupied electronic states (band structure below Fermi energy), and therefore, the gap was not observed for hole-doping system. The finite resolution and deep Fermi energy prevent the clear demonstration of all details in the entire valence bands. To account for all observed facts: (i) the absence of Weyl nodes along k_{\parallel} direction; (ii) the presence of Weyl nodes along k_z direction; (iii) gapped band structure (iv) the presence of field-induced VHS; we extend the minimal Weyl model to a more realistic model as illustrated in Fig. R15a. At a small energy range within the top of the valence band, 3D Weyl nodes are formed along k_z direction. For $k_z = 0$, the dispersion along k_{\parallel} direction should not exhibit crossing structures. At a large energy scale, all related bands bend downwards, forming a structure of so-called “Weyl semiconductor” [23,24]. The Weyl nodes along k_z direction in the realistic version of the model explain (i) the observation of Weyl nodes along $\Gamma - A$ in *SA* work [3]; (ii) the absence of the Weyl node along k_{\parallel} direction in *SA* and *PRL* work [3,19]; (iii) Weyl-related transport phenomena; (iv) metallic infrared response [14]; (v) the observation of VHS in our spectrum. On the other hand, the gapped structure in the band model explains: (i) the gap observed in the result of time-resolved APRES [19]; (ii) gapped features in infrared/transport experiment [19,26–28]; (iii) large energy of T_{β} in our spectrum. Most importantly, the core physics of this realistic band can still be described by the original toy model and explains the presence of 3D VHS.

Fig. R15 | Comparison between proposed band model and previous ARPES result. **a**, Proposed more realistic band structures along in-plane and out-of-plane directions. The conduction and valence bands are colored orange and light blue, respectively. The Weyl bands are highlighted by the black dashed box with red and blue curves denoting bands with opposite spins. The Fermi level of insulating and metallic samples are given by purple and blue dashed lines, respectively. **b**, ARPES result of the in-plane band structure in insulating sample (*PRL* work). Panel **i** and **ii** are the results of ARPES with/without Pump-probe. **c**, **d** ARPES result of in-plane and out-of-plane band structure in the metallic sample (*SA* work). The in-plane band structure is very similar to that in **b**, while the out-of-plane band structure exhibits the obvious band crossings. The finite resolution and deep Fermi energy might prevent the direct identification of all details in the entire valence band.

As suggested by the reviewer, this model also supports the sensitivity of transport property upon doping. In the light hole-doping case (with Fermi level denoted by purple dashed lines in Fig. R15a), the sample behaves as a slightly doped semiconductor or even an insulator. In the heavy hole-doping case (with Fermi level denoted by blue dashed lines in Fig. R15a), the system behaves as a (semi-)metal. The Weyl nodes below the Fermi level can contribute to unconventional transverse transport [17] and anomalous Hall effect [16,25] that have been explored in EuCd_2As_2 . The metal/insulator property might be tuned by different growth methods [3,26], control of impurity level [19], or the inclusion of the self-doping effect [19,29], but does not influence our finding since the observed optical transition and presence of VHS are not sensitive to the Fermi energy.

Compared to *PRL* paper, our contributions in the experimental part are to provide a larger spectral range and study the evolution of optical transition upon magnetization. *PRL* paper mainly focuses on the optical features around 700 meV. Here, we utilize the

magneto-near-infrared spectrum to exhibit other optical transitions ($T_\delta, T_\epsilon, T_\gamma$). We show that the system response is not limited around the ~ 700 meV (T_β). We theoretically predicted the near-infrared features based on the model and mid-infrared data. The consistency between prediction and experiments in the near-infrared regime helps to further validate the model. On the other hand, we demonstrate that the evolution of the optical transitions follows magnetization instead of magnetic fields, which allows us to identify the origin of the exchange interaction and exclude the Zeeman effect/cyclotron resonance. Following the “3-stage” evolution, the intensity variation agrees with the formation of 3D VHS.

In addition to the realistic model of EuCd_2As_2 , it might be worthwhile to emphasize that the key ingredient for accessing 3D VHS is the formation of the saddle ring structure. Therefore, it does not require the presence of low-energy Weyl nodes. For similar reasons, the formation of 3D VHS should not be influenced by the electronic structure at much higher energy (e.g., a large gap above VHS). On the other hand, our explanation of the observed transitions is not much influenced by the energy position of the Fermi level as well as the hole-doping extent in EuCd_2As_2 samples. The evolution of energy bands controlled by magnetization is the center for understanding the experimental phenomena. So, the controversy on metallic/insulating nature will not strongly influence the analysis of this manuscript.

We hope the realistic model is satisfying. It potentially contributes to the understanding of most current results in EuCd_2As_2 . For the above reason, we describe EuCd_2As_2 as a topological magnet regardless of its metallicity/insulativity.

Based on your comments, we have revised the manuscript as follows:

- *12. The schematic plot of the realistic band model has been provided as Fig. 5 in the main text.
- *13. We have introduced previous research (including *PRL* work) on metallic/insulating samples and explained the metallicity/insulativity based on the realistic version of the band model in the main text Page 12 Paragraph 2.
- *14. We add citation of *PRL* work [19] in main text as Ref. 80.
- *15. We have inverted the whole plot in Fig. 3b and Fig. 4a of the main text since the whole model is particle-hole symmetric. We explain that the inverted case helps to better align the model with ARPES results in the main text Page 8 Paragraph 3.
- *16. The description of upper and lower flat bands has been modified throughout the main text.

*17. We have clarified the contribution of our magneto-infrared study to the understanding of EuCd_2As_2 compared to *PRL* work in Supplementary Section XI.

*18. We have discussed the consistency between the proposed model and previous ARPES/optical results in the main text Page 11 Paragraph 3 and Page 12 Paragraph 2.

Comment 6:

In fact, this interpretation of the spectra do not contradict with an earlier infrared spectroscopy data (PRB 94, 045112 (2016)), where a metallic sample is used similar to the common literature (presumably also similar to the one in this work). Here a very small contribution of the Drude component along with the high energy absorption with an onset at high energy regime can very well be interpreted as a slightly doped semiconducting behavior. Overall, I believe the current compound is not very trivial and authors should take into account the possible impurity and self-doping effect of the compound.

Response:

We appreciate the reviewer bringing up the impurity and self-doping effect in EuCd_2As_2 . These effects significantly influence the transport property of the compound upon tuning the Fermi energy. *Wang et al.* report the infrared spectroscopy study of EuCd_2As_2 in *Phys. Rev. B* 94, 045112 (2016) [14] (refers to *PRB* work, hereafter), which indicates the metallic phase in EuCd_2As_2 with low carrier density and electron-phonon interaction. It agrees with our realistic version of the band model as shown in Fig. R15a and discussed in response to Comment #5. The transport property of EuCd_2As_2 is sensitive to the self-doping level. The spectral phenomena in *PRB* work can be explained by the proposed band structure with the Fermi level denoted by the blue dashed line (Fig. R15a). Notably, in their metallic sample, the hole-doping level is expected to be heavier than that in the insulating sample, but not heavy enough to render the sample a complete metal. Therefore, as the reviewer mentioned, the spectral features observed in the *PRB* work are similar to doped semiconductors. Our sample shows insulating behavior in resistivity, indicating a comparatively lower self-doping level.

It is worth emphasizing again that the observed optical transitions in our spectrum are excitations between the VHS and other electronic states away from the Fermi energy. These optical transitions mainly reflect the field evolution of band structures so that they are insensitive to the Fermi level. The presence of the Weyl semimetal phase is not the prerequisite for the discussed VHS physics. The more realistic version of the model helps to understand both the semiconductors nature and Weyl-related ARPES/transport

results. The Fermi energy determined by the self-doping effect and impurity condition are indeed important to help understand the distinct resistivity behavior of the compound.

Based on your comments, we have revised the manuscript as follows:

*19. We have provided the realistic band model and explained the correlation between experimental phenomena and the self-doping effect in the main text the main text Page 12 Paragraph 2. The *PRB* work is also cited and discussed in this paragraph.

We would like to sincerely appreciate Reviewer #3 for crucial and insightful comments on our manuscript, especially on the semiconductor gap reported in recent work and spectral features evolution around the critical field. Following reviewer's suggestion, we build a more realistic model that potentially gives a universal picture for understanding our results and previous research based on different samples and various techniques. With additional analysis of spectra and model calculations, our conclusions are further consolidated. We hope that the revised manuscript could address all your concerns. Thank you very much.

References

- [1] N. F. Q. Yuan, H. Isobe, and L. Fu, *Magic of High-Order van Hove Singularity*, Nat Commun **10**, 5769 (2019).
- [2] N. F. Q. Yuan and L. Fu, *Classification of Critical Points in Energy Bands Based on Topology, Scaling, and Symmetry*, Phys. Rev. B **101**, 125120 (2020).
- [3] J.-Z. Ma, S. M. Nie, C. J. Yi, J. Jandke, T. Shang, M. Y. Yao, M. Naamneh, L. Q. Yan, Y. Sun, A. Chikina, V. N. Strocov, M. Medarde, M. Song, Y.-M. Xiong, G. Xu, W. Wulfhekel, J. Mesot, M. Reticcioli, C. Franchini, C. Mudry, M. Müller, Y. G. Shi, T. Qian, H. Ding, and M. Shi, *Spin Fluctuation Induced Weyl Semimetal State in the Paramagnetic Phase of EuCd_2As_2* , Sci. Adv. **5**, eaaw4718 (2019).
- [4] G. Chen, A. L. Sharpe, P. Gallagher, I. T. Rosen, E. J. Fox, L. Jiang, B. Lyu, H. Li, K. Watanabe, T. Taniguchi, J. Jung, Z. Shi, D. Goldhaber-Gordon, Y. Zhang, and F. Wang, *Signatures of Tunable Superconductivity in a Trilayer Graphene Moiré Superlattice*, Nature **572**, 7768 (2019).
- [5] L. Ohnoutek, M. Hakl, M. Veis, B. A. Piot, C. Faugeras, G. Martinez, M. V. Yakushev, R. W. Martin, Č. Drašar, A. Materna, G. Strzelecka, A. Hruban, M. Potemski, and M. Orlita, *Strong Interband Faraday Rotation in 3D Topological Insulator Bi_2Se_3* , Sci. Rep. **6**, 19087 (2016).
- [6] H. Krenn, W. Herbst, H. Pascher, Y. Ueta, G. Springholz, and G. Bauer, *Interband*

- Faraday and Kerr Rotation and Magnetization of $Pb_{1-x}Eu_xTe$ in the Concentration Range $0 < x \leq 1$* , Phys. Rev. B **60**, 8117 (1999).
- [7] D. U. Bartholomew, J. K. Furdyna, and A. K. Ramdas, *Interband Faraday Rotation in Diluted Magnetic Semiconductors: $Zn_{1-x}Mn_xTe$ and $Cd_{1-x}Mn_xTe$* , Phys. Rev. B **34**, 6943 (1986).
- [8] M. Dressel and G. Grüner, *Electrodynamics of Solids: Optical Properties of Electrons in Matter* (Cambridge University Press, 2002).
- [9] R. Yang, T. Zhang, L. Zhou, Y. Dai, Z. Liao, H. Weng, and X. Qiu, *Magnetization-Induced Band Shift in Ferromagnetic Weyl Semimetal $Co_3Sn_2S_2$* , Phys. Rev. Lett. **124**, 077403 (2020).
- [10] B. Xu, Y. M. Dai, L. X. Zhao, K. Wang, R. Yang, W. Zhang, J. Y. Liu, H. Xiao, G. F. Chen, A. J. Taylor, D. A. Yarotski, R. P. Prasankumar, and X. G. Qiu, *Optical Spectroscopy of the Weyl Semimetal TaAs*, Phys. Rev. B **93**, 121110 (2016).
- [11] D. Santos-Cottin, J. Wyzula, F. Le Mardelé, I. Crassee, E. Martino, J. Novák, G. Eguchi, Z. Rukelj, M. Novak, M. Orlita, and A. Akrap, *Addressing Shape and Extent of Weyl Cones in TaAs by Landau Level Spectroscopy*, Phys. Rev. B **105**, L081114 (2022).
- [12] S. Polatkan, M. O. Goerbig, J. Wyzula, R. Kemmler, L. Z. Maulana, B. A. Piot, I. Crassee, A. Akrap, C. Shekhar, C. Felser, M. Dressel, A. V. Pronin, and M. Orlita, *Magneto-Optics of a Weyl Semimetal beyond the Conical Band Approximation: Case Study of TaP*, Phys. Rev. Lett. **124**, 176402 (2020).
- [13] Y. Jiang, Z. Dun, S. Moon, H. Zhou, M. Koshino, D. Smirnov, and Z. Jiang, *Landau Quantization in Coupled Weyl Points: A Case Study of Semimetal NbP*, Nano Lett. **18**, 7726 (2018).
- [14] H. P. Wang, D. S. Wu, Y. G. Shi, and N. L. Wang, *Anisotropic Transport and Optical Spectroscopy Study on Antiferromagnetic Triangular Lattice $EuCd_2As_2$: An Interplay between Magnetism and Charge Transport Properties*, Phys. Rev. B **94**, 045112 (2016).
- [15] J. Ma, H. Wang, S. Nie, C. Yi, Y. Xu, H. Li, J. Jandke, W. Wulfhekel, Y. Huang, D. West, P. Richard, A. Chikina, V. N. Strocov, J. Mesot, H. Weng, S. Zhang, Y. Shi, T. Qian, M. Shi, and H. Ding, *Emergence of Nontrivial Low-Energy Dirac Fermions in Antiferromagnetic $EuCd_2As_2$* , Adv. Mater. **32**, 1907565 (2020).
- [16] S. Roychowdhury, M. Yao, K. Samanta, S. Bae, D. Chen, S. Ju, A. Raghavan, N. Kumar, P. Constantinou, S. N. Guin, N. C. Plumb, M. Romanelli, H. Borrmann, M. G. Vergniory, V. N. Strocov, V. Madhavan, C. Shekhar, and C. Felser, *Anomalous Hall Conductivity and Nernst Effect of the Ideal Weyl Semimetallic Ferromagnet $EuCd_2As_2$* , Adv. Sci. **10**, 2207121 (2023).

- [17] Y. Xu, L. Das, J. Z. Ma, C. J. Yi, S. M. Nie, Y. G. Shi, A. Tiwari, S. S. Tsirkin, T. Neupert, M. Medarde, M. Shi, J. Chang, and T. Shang, *Unconventional Transverse Transport above and below the Magnetic Transition Temperature in Weyl Semimetal EuCd_2As_2* , Phys. Rev. Lett. **126**, 076602 (2021).
- [18] X. Cao, J.-X. Yu, P. Leng, C. Yi, X. Chen, Y. Yang, S. Liu, L. Kong, Z. Li, X. Dong, Y. Shi, M. Bibes, R. Peng, J. Zang, and F. Xiu, *Giant Nonlinear Anomalous Hall Effect Induced by Spin-Dependent Band Structure Evolution*, Phys. Rev. Res. **4**, 023100 (2022).
- [19] D. Santos-Cottin, I. Mohelský, J. Wyzula, F. Le Mardelé, I. Kapon, S. Nasrallah, N. Barišić, I. Živković, J. R. Soh, F. Guo, K. Rigaux, M. Puppín, J. H. Dil, B. Gudac, Z. Rukelj, M. Novak, A. B. Kuzmenko, C. C. Homes, T. Dietl, M. Orlita, and A. Akrap, *EuCd_2As_2 : A Magnetic Semiconductor*, Phys. Rev. Lett. **131**, 186704 (2023).
- [20] L.-L. Wang, N. H. Jo, B. Kuthanazhi, Y. Wu, R. J. McQueeney, A. Kaminski, and P. C. Canfield, *Single Pair of Weyl Fermions in the Half-Metallic Semimetal EuCd_2As_2* , Phys. Rev. B **99**, 245147 (2019).
- [21] G. Hua, S. Nie, Z. Song, R. Yu, G. Xu, and K. Yao, *Dirac Semimetal in Type-IV Magnetic Space Groups*, Phys. Rev. B **98**, 201116 (2018).
- [22] M. C. Rahn, J.-R. Soh, S. Francoual, L. S. I. Veiga, J. Stremper, J. Mardegan, D. Y. Yan, Y. F. Guo, Y. G. Shi, and A. T. Boothroyd, *Coupling of Magnetic Order and Charge Transport in the Candidate Dirac Semimetal EuCd_2As_2* , Phys. Rev. B **97**, 214422 (2018).
- [23] G. Qiu, C. Niu, Y. Wang, M. Si, Z. Zhang, W. Wu, and P. D. Ye, *Quantum Hall Effect of Weyl Fermions in N-Type Semiconducting Tellurene*, Nat. Nanotechnol. **15**, 585 (2020).
- [24] J. Chen, T. Zhang, J. Wang, L. Xu, Z. Lin, J. Liu, C. Wang, N. Zhang, S. P. Lau, W. Zhang, M. Chhowalla, and Y. Chai, *Topological Phase Change Transistors Based on Tellurium Weyl Semiconductor*, Science Advances **8**, eabn3837 (2022).
- [25] J.-R. Soh, F. de Juan, M. G. Vergniory, N. B. M. Schröter, M. C. Rahn, D. Y. Yan, J. Jiang, M. Bristow, P. A. Reiss, J. N. Blandy, Y. F. Guo, Y. G. Shi, T. K. Kim, A. McCollam, S. H. Simon, Y. Chen, A. I. Coldea, and A. T. Boothroyd, *Ideal Weyl Semimetal Induced by Magnetic Exchange*, Phys. Rev. B **100**, 201102 (2019).
- [26] X. Chen, S. Wang, J. Wang, C. An, Y. Zhou, Z. Chen, X. Zhu, Y. Zhou, Z. Yang, and M. Tian, *Temperature-Pressure Phase Diagram of the Intrinsically Insulating Topological Antiferromagnet EuCd_2As_2* , Phys. Rev. B **107**, L241106 (2023).
- [27] Y. Wang, J. Ma, J. Yuan, W. Wu, Y. Zhang, Y. Mou, J. Gu, P. Cheng, W. Shi, X. Yuan, J. Zhang, Y. Guo, and C. Zhang, *Absence of Metallicity and Bias-Dependent*

Resistivity in Low-Carrier-Density EuCd₂As₂, arXiv:2311.11515.

- [28] M. Wu, R. Yang, X. Zhu, Y. Ren, A. Qian, Y. Xie, C. Yue, Y. Nie, X. Yuan, N. Wang, D. Tu, D. Li, Y. Han, Z. Wang, Y. Dai, G. Zheng, J. Zhou, W. Ning, X. Qiu, and M. Tian, *Surface Skyrmions and Dual Topological Hall Effect in Antiferromagnetic Topological Insulator EuCd₂As₂*, arXiv:2311.15835.
- [29] Y. Wang, C. Li, T. Miao, S. Zhang, Y. Li, L. Zhou, M. Yang, C. Yin, Y. Cai, C. Song, H. Luo, H. Chen, H. Mao, L. Zhao, H. Deng, Y. Sun, C. Zhu, F. Zhang, F. Yang, Z. Wang, S. Zhang, Q. Peng, S. Pan, Y. Shi, H. Weng, T. Xiang, Z. Xu, and X. J. Zhou, *Giant and Reversible Electronic Structure Evolution in a Magnetic Topological Material EuCd₂As₂*, Phys. Rev. B **106**, 085134 (2022).

REVIEWERS' COMMENTS

Reviewer #1 (Remarks to the Author):

I think my questions are well addressed in a satisfactory way. I recommend the publication of the revised manuscript on *Nature Communications*.

Reviewer #2 (Remarks to the Author):

In the previous round, I pointed out what was unclear and requested several things to be modified. In particular, it was not clear how they extracted the spectral dispersion from the color contrast image. In the new version, they plot raw spectra and demonstrate how they extracted the dispersions. Thanks to that, it has now become rather clearer that their data quality was high enough to extract information leading to their claims. In addition, I pointed out that the energy location of 4f flat bands does not match what they claim for their "flat band". In reply, they argue that the flat bands they discuss are not for 4f bands but "partially" flat bands coming from other orbitals. By adding this argument to the revised manuscript, confusion for the readers has been mitigated.

In conclusion, the authors satisfactorily replied to my concerns and also addressed all my requests in the revised manuscript. I think that the authors' claims are now well justified with improved presentations. The realization of the 3D vHS by manipulating the Weyl state with a magnetic field is very interesting and has a potentially high impact on the community. Therefore, I recommend the publication of this revised version of the manuscript to *Nature Communications*.

Reviewer #3 (Remarks to the Author):

Authors revised the manuscript based on the comments from all the reviewers. I believe the revision and the reply to the comments are extensive. Now, we can also see the raw-spectra, where indeed allow one to follow the mentioned features more easily. The explanation regarding the proposed band structure is also helpful. Therefore, I recommend the publication of this manuscript in *Nature Communications*. I believe the comparison to the other magneto-optics work also provides important points to be considered in general (beyond this specific compound).

Response to Reviewers' Comments

Reviewer 1

Comment:

I think my questions are well addressed in a satisfactory way. I recommend the publication of the revised manuscript on Nature Communications.

Response:

We sincerely thank Reviewer#1 for the time and effort dedicated to reviewing and improving our manuscript.

Reviewer 2

Comment:

In the previous round, I pointed out what was unclear and requested several things to be modified. In particular, it was not clear how they extracted the spectral dispersion from the color contrast image. In the new version, they plot raw spectra and demonstrate how they extracted the dispersions. Thanks to that, it has now become rather clearer that their data quality was high enough to extract information leading to their claims. In addition, I pointed out that the energy location of 4f flat bands does not match what they claim for their "flat band". In reply, they argue that the flat bands they discuss are not for 4f bands but "partially" flat bands coming from other orbitals. By adding this argument to the revised manuscript, confusion for the readers has been mitigated.

In conclusion, the authors satisfactorily replied to my concerns and also addressed all my requests in the revised manuscript. I think that the authors' claims are now well justified with improved presentations. The realization of the 3D vHS by manipulating the Weyl state with a magnetic field is very interesting and has a potentially high impact on the community. Therefore, I recommend the publication of this revised version of the manuscript to Nature Communications.

Response:

We are glad for addressing all of Reviewer#2's concerns and sincerely thank Reviewer#2 for giving an encouraging comment on our work.

Reviewer 3

Comment:

Authors revised the manuscript based on the comments from all the reviewers. I believe the revision and the reply to the comments are extensive. Now, we can also see the raw-

spectra, where indeed allow one to follow the mentioned features more easily. The explanation regarding the proposed band structure is also helpful. Therefore, I recommend the publication of this manuscript in Nature Communications. I believe the comparison to the other magneto-optics work also provides important points to be considered in general (beyond this specific compound).

Response:

We are glad for addressing all of Reviewer#3's concerns and sincerely appreciate Reviewer#3 for insightful suggestions on our manuscript.